# Understanding Model Reprogramming for CLIP via Decoupling Visual Prompts

**Chengyi Cai** [* 1]  **Zesheng Ye** [* 1]  **Lei Feng** [2 3]  **Jianzhong Qi** [1]  **Feng Liu** [1]

## Abstract

Model reprogramming adapts pretrained models to downstream tasks by modifying only the input and output spaces. *Visual reprogramming* (VR) is one instance for vision tasks that adds a trainable noise pattern (i.e., a visual prompt) to input images to facilitate downstream classification. The existing VR approaches for CLIP train a single visual prompt using all descriptions of different downstream classes. However, the limited learning capacity may result in (1) a failure to capture diverse aspects of the descriptions (e.g., shape, color, and texture), and (2) a possible bias toward less informative attributes that do not help distinguish between classes. In this paper, we introduce a decoupling-and-reweighting framework. Our *decoupled visual prompts* (DVP) are optimized using descriptions grouped by explicit **causes** (DVP-cse) or unsupervised **clusters** (DVP-cls). Then, we integrate the outputs of these visual prompts with a *probabilistic reweighting matrix* (PRM) that measures their contributions to each downstream class. Theoretically, DVP lowers the empirical risk bound. Experimentally, DVP outperforms baselines on average across 11 downstream datasets. Notably, the DVP-PRM integration enables insights into how individual visual prompts influence classification decisions, providing a probabilistic framework for understanding reprogramming. Our code is available at https://github.com/tmlr-group/DecoupledVP

## 1. Introduction

Model reprogramming (Vinod et al., 2020; Chen, 2024; Hung et al., 2023) is shown to be an effective approach for

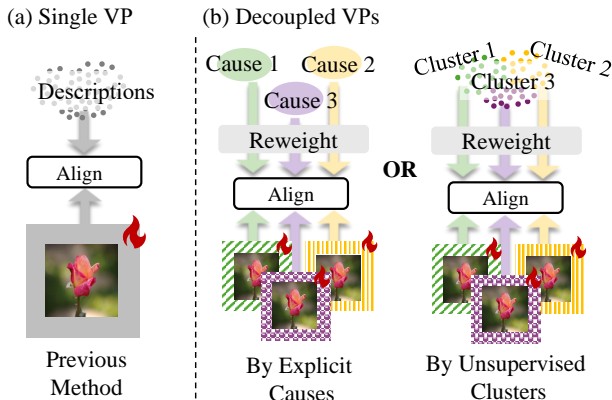

(a) Single VP    (b) Decoupled VPs

Descriptions

Align

Previous Method

Cause 1   Cause 2   Cause 3   Reweight   Align   By Explicit Causes

Cluster 1   Cluster 2   Cluster 3   Reweight   Align   By Unsupervised Clusters

OR

*Figure 1.* Difference between (a) existing VR methods that train a single VP for all descriptions, and (b) our DVP that trains decoupled-and-reweighted VPs that are optimized using descriptions grouped by explicit causes or unsupervised clusters. Learnable parameters are marked with 'fire's.

adapting models pretrained on abundant data to a specific task by modifying inputs and outputs without altering the model's core architecture or retraining. Among these methods, *visual reprogramming* (VR) (Cai et al., 2024b;a; Tsao et al., 2024; Chen et al., 2023), also known as adversarial reprogramming (Elsayed et al., 2018; Tsai et al., 2020), aims to repurpose pretrained models for downstream image classification tasks by adding trainable noise patterns to the input images. VR for *vision-language models* (VLMs) (Cai et al., 2025) trains a *single* noise pattern, also known as a *visual prompt* (VP), to align input images with (i.e., bring the visual features closer to texture features) *all* descriptions of different downstream classes, as shown in Figure 1(a).

However, attribute descriptions (Cai et al., 2025) are complex and diverse, while the learning capacity of a single VP may be limited, potentially insufficient to handle all image-description pairs. This may result in the VP successfully capturing descriptions in certain aspects while failing in others. Additionally, the VP may unreasonably tend to optimize less informative descriptions that, while easier to match with images (i.e., having a higher cosine similarity), exhibit low discriminability between classes. Figure 2 shows an example of a single VP pattern trained on the OxfordFlowers (Nilsback & Zisserman, 2008) dataset for all attributes (Cai et al., 2025), with sample descriptions shown

*Equal contribution [1]School of Computing and Information Systems, The University of Melbourne [2]School of Computer Science and Engineering, Southeast University [3]Idealism Technology (Beijing). Correspondence to: Feng Liu <fengliu.ml@gmail.com>.

*Proceedings of the 42nd International Conference on Machine Learning*, Vancouver, Canada. PMLR 267, 2025. Copyright 2025 by the author(s).

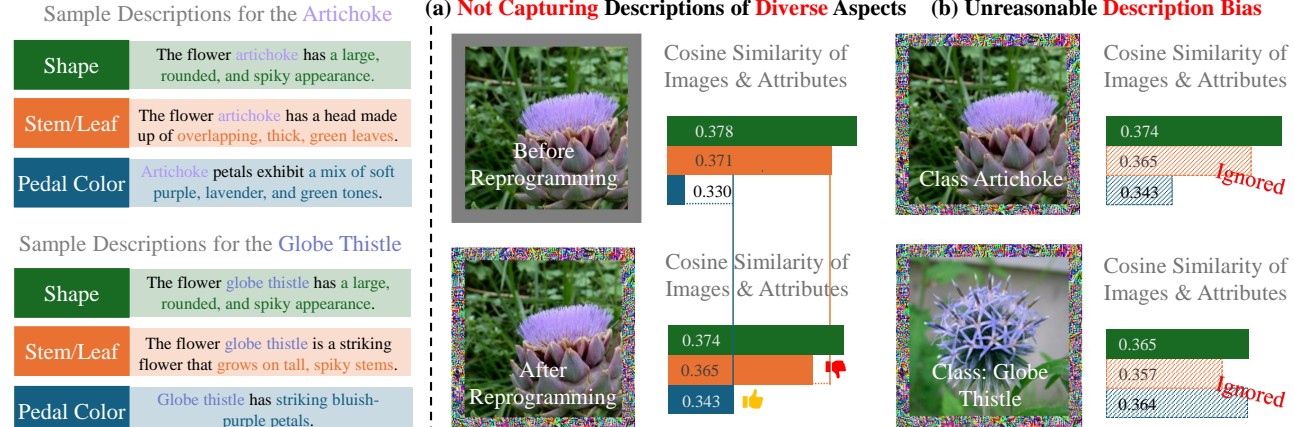

*Figure 2.* Drawbacks of training a single VP with all attribute descriptions. Sample descriptions of different aspects for 'artichoke' and 'globe thistle' are shown on the left, while recorded cosine similarities of images and descriptions are on the right. Figure (a) illustrates its inability to capture descriptions of diverse aspects, and Figure (b) reveals its unreasonable bias towards less informative descriptions.

on the left. Three attributes of different aspects–'shape', 'stem/leaf' and 'petal color'–are listed for class 'artichoke' and 'globe thistle'. Figure 2(a) shows the cosine similarity between attribute descriptions and the image 'artichoke' before and after reprogramming using the VP pattern. It can be observed that although adding VP enlarges the similarity between the input image and the description regarding 'petal color', the description about 'stem/leaf'–also being an important aspect for distinguishing artichokes–has, conversely, become more alien. This illustrates the inability of a single VP to capture descriptions of diverse aspects. Figure 2(b) shows the cosine similarity between the reprogrammed image 'artichoke' or 'globe thistle' and their corresponding descriptions. Intuitively, both flowers share the same 'shape' (i.e., large, round and spiky), while aspects like 'stem/leaf' and 'petal color' may be more informative for distinguishing between these two classes. However, as 'shape' descriptions have a higher similarity with the images, the VP learning process favors them. It highlights how using a single VP may exhibit a bias towards less informative descriptions.

In this paper, we first formulate the problem and then introduce a decoupling-and-reweighting learning framework (as shown in Figure 1(b)) to avoid the above drawbacks of a single VP. Under our framework, decoupled VPs will be optimized for different description partitions, and the importance of descriptions is distinguished through reweighting (see Section 3 for details).

In Section 4, for the decoupling step, we propose *decoupled visual prompts* (DVP) that are optimized using descriptions grouped by explicit *causes* (DVP-cse) or unsupervised *clusters* (DVP-cls). Explicit causes can be determined by querying a *large language model* (LLM) while unsupervised clusters can be resolved with K-means. For the reweighting step, the output of DVP will be integrated using a *proba-*

*bility reweighting matrix* (PRM) which is optimized during learning through *maximum likelihood estimation* (MLE). We also show that DVP improves the results by lowering the empirical risk bound.

Section 5 shows the application of DVP to 11 commonly used downstream datasets and four CLIP backbones, demonstrating its effectiveness. The parameter analysis, ablation experiments, and independence tests further validate the rationality of DVP. Remarkably, the visualization results of PRM reflect how individual VPs influence classification decisions, reflecting their different roles and contributions across various downstream classes.

In conclusion, both theoretical analysis and experimental results verify the soundness of DVP. The DVP-PRM integration also offers a probabilistic framework for understanding model reprogramming by exploring individual VPs.

## 2. Related Works

**Model Reprogramming.** Model reprogramming (Chen, 2024) is a supervised finetuning approach that achieves transfer to downstream tasks while preserving the integrity of pretrained models, by solely modifying the input and output spaces. It has demonstrated promising applications across pretrained vision (Chen et al., 2023; Tsai et al., 2020; Cai et al., 2024a; Jin et al., 2025), graph (Jing et al., 2023), acoustic (Yang et al., 2021; 2023; Hung et al., 2023; Yen et al., 2023), and language models (Hambardzumyan et al., 2021; Vinod et al., 2020; Jin et al., 2024). Comparatively, it not only ensures the completeness of the pretrained model, preventing catastrophic forgetting (Kirkpatrick et al., 2017) but also enables architecture-agnostic transfer to downstream tasks with minimal parameter adjustments. The robustness of model reprogramming has recently been in-

vestigated (Chen et al., 2025; Zhou et al., 2025).

Input VR (Cai et al., 2024b;a; Chen et al., 2023) is a type of model reprogramming, primarily used to apply pretrained models to downstream image classification tasks by adding trainable noise patterns to the input images. The main strategies include padding trainable parameters around the images (Chen et al., 2023; Tsai et al., 2020; Tsao et al., 2024) or adding watermarks (Bahng et al., 2022; Oh et al., 2023) to rescaled images, which has been successfully applied to both unimodal image classifiers (Chen et al., 2023; Cai et al., 2024a) and VLMs (Oh et al., 2023; Zhang et al., 2024).

**Prompt Learning.** Slightly different from model reprogramming, prompts (Jia et al., 2022) serve as learnable weights that can be attached to any position of a pretrained model. Therefore, prompt design often needs to consider the specific architecture of the pretrained model. Prompts can be added as text prompts (Zhou et al., 2022b;a) among input words, as VPs (Chen et al., 2023; Oh et al., 2023; Tsao et al., 2024) overlaying input images, as token prompts (Wang et al., 2023) within self-attention layers, or as mappings (Khattak et al., 2023) between modalities to connect the text and image embeddings.

Specifically, adding VPs to input images for reusing a pretrained VLM to downstream classification tasks is equivalent to VR methods for VLMs. Among recent research, VP (Bahng et al., 2022) overlays watermarking patterns on images, AR (Tsai et al., 2020; Chen et al., 2023) pads patterns around images, BlackVIP (Oh et al., 2023) aims at black-box transfer learning, DAM (Huang et al., 2023) partitions the image set for divide-and-conquer training, and AttrVR (Cai et al., 2025) introduces attributes to better align images with texts. These existing methods optimize single VP patterns for all descriptions. Our proposed DVP strives to train VPs with diverse roles and contributions, enhancing both learning capability and interpretability.

## 3. Preliminaries and Insights

We follow standard protocols for VR on CLIP (Chen et al., 2023; Cai et al., 2025). See Appendix D.1 for a setup comparison between our problem and text prompt tuning (Zhou et al., 2022a), and Appendix D.2 for a summary of notation.

**CLIP-based Image Classification.** CLIP (Radford et al., 2021) is a pre-trained VLM with an image encoder $f_{\mathrm{img}}$ : $\mathcal{X}^{\mathrm{S}} \to \mathcal{Z}$ that projects input images from a $d_{\mathrm{S}}$-dimensional image space $\mathcal{X}^{\mathrm{S}} \subseteq \mathbb{R}^{d_{\mathrm{S}}}$ to embedding space $\mathcal{Z}$, and a text encoder $f_{\mathrm{txt}} : \mathcal{V} \to \mathcal{Z}$ that projects texts $V \in \mathcal{V}$ to the same embedding space $\mathcal{Z}$. The similarity between an image $x^{\mathrm{S}} \in \mathcal{X}^{\mathrm{S}}$ and a text $V \in \mathcal{V}$ is:

$$f_{\mathrm{clip}}(x^{\mathrm{S}}, V) = \cos\left(f_{\mathrm{img}}(x^{\mathrm{S}}), f_{\mathrm{txt}}(V)\right)/\tau,$$

where $\cos(\cdot, \cdot)$ is cosine similarity and $\tau$ is the temperature.

When CLIP is used for a downstream classification task defined over $\mathcal{X}^{\mathrm{T}} \times \mathcal{Y}^{\mathrm{T}}$, where $\mathcal{X}^{\mathrm{T}} \subseteq \mathbb{R}^{d_{\mathrm{T}}}$ and $\mathcal{Y}^{\mathrm{T}}$ respectively represent the $d_{\mathrm{T}}$-dimensional image space and the label space, we often use textual descriptions $\mathcal{A} \subseteq \mathcal{V}$ to represent class labels as texts that CLIP can process. Often, a description can be implemented as a template with placeholders, e.g., "This is a photo of [Class Name]." (Radford et al., 2021) or a set of attributes for the class, e.g., "[Class Name] is [Attributes]." (Cai et al., 2025). Let $\mathcal{A}(y^{\mathrm{T}})$ be the subset of $m$ descriptions for a class $y^{\mathrm{T}} \in \mathcal{Y}^{\mathrm{T}}$, with description sets for different classes being disjoint, i.e., $\mathcal{A}(y_i^{\mathrm{T}}) \cap \mathcal{A}(y_j^{\mathrm{T}}) = \varnothing$ for $y_i^{\mathrm{T}} \neq y_j^{\mathrm{T}}$, and $\mathcal{A} = \bigcup_{y^{\mathrm{T}} \in \mathcal{Y}^{\mathrm{T}}} \mathcal{A}(y^{\mathrm{T}})$.

Let $f_{\mathrm{logits}} : \mathcal{X}^{\mathrm{T}} \times \mathcal{A} \to \mathbb{R}^{|\mathcal{Y}^{\mathrm{T}}|}$ be the logit vector over the label space $\mathcal{Y}^{\mathrm{T}}$. If an input image $x^{\mathrm{T}} \in \mathcal{X}^{\mathrm{T}}$ matches CLIP's input dimension ($d_{\mathrm{T}} = d_{\mathrm{S}}$), the logit for class $y^{\mathrm{T}}$ is

$$[f_{\mathrm{logits}}(x^{\mathrm{T}}; \mathcal{A})]_{y^{\mathrm{T}}} = \mathrm{agg}_{a \in \mathcal{A}(y^{\mathrm{T}})} \left( f_{\mathrm{clip}}(x^{\mathrm{T}}, a) \right),$$

where $a$ is a description and $\mathrm{agg}(\cdot)$ is an aggregation function, typically $\max(\cdot)$ or $\mathrm{avg}(\cdot)$.

**VR on CLIP.** For mismatched downstream input shapes, i.e., $d_{\mathrm{T}} \neq d_{\mathrm{S}}$, standard input VR (Cai et al., 2024b) uses an input transform $f_{\mathrm{in}} : \mathbb{R}^{d_{\mathrm{T}}} \to \mathbb{R}^{d_{\mathrm{S}}}$ defined as $f_{\mathrm{in}}(x^{\mathrm{T}}|\delta) \triangleq \mathrm{pad}(x^{\mathrm{T}}) + \delta$, where $\mathrm{pad}(\cdot)$ function pads zeros around the images and VR patterns $\delta \in \mathbb{R}^{d_{\mathrm{S}}}$ is a trainable visual prompt. The logit for a class $y^{\mathrm{T}} \in \mathcal{Y}^{\mathrm{T}}$ given $x^{\mathrm{T}}$ and prompt $\delta$ is:

$$\left[f_{\mathrm{logits}}^{\mathrm{vr}}(x^{\mathrm{T}}; \delta, \mathcal{A})\right]_{y^{\mathrm{T}}} = \mathrm{agg}_{a \in \mathcal{A}(y^{\mathrm{T}})} \left( f_{\mathrm{clip}}(f_{\mathrm{in}}(x^{\mathrm{T}}|\delta), a) \right). \quad (1)$$

Then, the normalized probability will be obtained by

$$p_{\mathrm{vr}}(y^{\mathrm{T}}|x^{\mathrm{T}}; \delta, \mathcal{A}) = \frac{\exp([f_{\mathrm{logits}}^{\mathrm{vr}}(x^{\mathrm{T}}; \delta, \mathcal{A})]_{y^{\mathrm{T}}})}{\sum_{y' \in \mathcal{Y}^{\mathrm{T}}} \exp([f_{\mathrm{logits}}^{\mathrm{vr}}(x^{\mathrm{T}}; \delta, \mathcal{A})]_{y'})}. \quad (2)$$

The parameter $\delta$ in VR can be learned by minimizing the negative log-likelihood, computed over a downstream training set $\mathcal{D} = \{(x_j^{\mathrm{T}}, y_j^{\mathrm{T}})\}_{j=1}^{N} \overset{\mathrm{i.i.d}}{\sim} \mathcal{X}^{\mathrm{T}} \times \mathcal{Y}^{\mathrm{T}}$ as,

$$\delta^* = \arg\min_{\delta} -\frac{1}{N} \sum_{j=1}^{N} \left[ \log p_{\mathrm{vr}}(y_j^{\mathrm{T}}|x_j^{\mathrm{T}}; \delta, \mathcal{A}) \right], \quad (3)$$

where $N = n \times |\mathcal{Y}^{\mathrm{T}}|$ is the size of training dataset with $n$ samples per class.

**Representing in the Matrix Form.** To rewrite Eq. (1) in the matrix form, for certain input image $x^{\mathrm{T}}$, we can define a row vector $M_a = [\tilde{a}_1, \tilde{a}_2, \ldots, \tilde{a}_{|\mathcal{A}|}]$ being the CLIP output $f_{\mathrm{clip}}(f_{\mathrm{in}}(x^{\mathrm{T}}|\delta), \cdot)$ across different attribute descriptions and another row vector $M_y = [\tilde{y}_1, \tilde{y}_2, \ldots, \tilde{y}_{|\mathcal{Y}^{\mathrm{T}}|}]$ being the logits $f_{\mathrm{logits}}^{\mathrm{vr}}(x^{\mathrm{T}}; \delta, \mathcal{A})$ across different downstream labels. Then they can be related by a reweighting matrix $\omega \in \mathbb{R}^{|\mathcal{A}| \times |\mathcal{Y}^{\mathrm{T}}|}$ that $M_a \cdot \omega = M_y$:

$$\begin{bmatrix} \tilde{a}_1 \\ \tilde{a}_2 \\ \cdots \\ \tilde{a}_{|\mathcal{A}|} \end{bmatrix}^{\top} \cdot \begin{bmatrix} \omega_{1,1} & \cdots & \omega_{1,|\mathcal{Y}^{\mathrm{T}}|} \\ \vdots & \ddots & \vdots \\ \omega_{|\mathcal{A}|,1} & \cdots & \omega_{|\mathcal{A}|,|\mathcal{Y}^{\mathrm{T}}|} \end{bmatrix} = \begin{bmatrix} \tilde{y}_1 \\ \tilde{y}_2 \\ \cdots \\ \tilde{y}_{|\mathcal{Y}^{\mathrm{T}}|} \end{bmatrix}^{\top}, \quad (4)$$

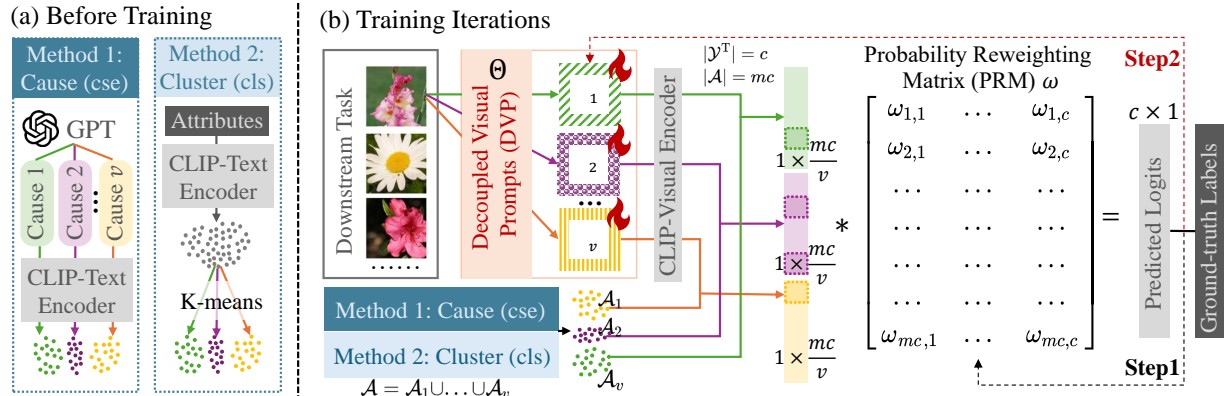

Figure 3. Pipeline of DVP. Before training, the description partitions $\mathcal{A}_1, \ldots, \mathcal{A}_v$ are determined–either by generating descriptions using an LLM based on different *causes* (DVP-cse) or by *clustering* existing attribute descriptions (DVP-cls). During training, VPs are trained for their corresponding description partitions separately, after being integrated and reweighted by PRM $\omega$. In each iteration, $\omega$ is first estimated, followed by the update of VPs $\Delta = \{\delta_1, \ldots, \delta_v\}$, w.r.t. negative log-likelihood (i.e., cross-entropy loss in classification tasks).

where each element $\omega_{p,q}$ in $\omega$ represents the contribution from $\tilde{a}_p$ (i.e., CLIP output of the $p$-th attribute description) to $\tilde{y}_q$ (i.e., logit output for the $q$-th label). Here, Eq. (4) implies that $\text{agg}(\cdot)$ implements a *fixed reweighting* scheme. Concretely, if $\max(\cdot)$ is used in Eq. (1), then $\omega \in \{0, 1\}^{|\mathcal{A}| \times |\mathcal{Y}^T|}$. $\omega_{p,q}$ is set to be 1 only when $a_p$ is the maximum CLIP output for the $q$-th class's attribute description, otherwise $\omega_{p,q} = 0$. If $\text{avg}(\cdot)$ is used, then $\omega \in \{0, \frac{1}{m}\}^{|\mathcal{A}| \times |\mathcal{Y}^T|}$, and $\omega_{p,q} = 1/m$ only when the $p$-th attribute is a description related to the $q$-th class.

Then we can replace $\text{agg}(\cdot)$ with a fixed $\omega^{\text{fix}}$ and rewrite the logits of label $y_q$ in Eq. (1) as $[f_{\text{logits}}^{\text{vr}}(x^T; \delta, \omega^{\text{fix}}, \mathcal{A})]_{y_q} = \sum_{a_p \in \mathcal{A}} \omega_{p,q}^{\text{fix}} \cdot f_{\text{clip}}(f_{\text{in}}(x^T|\delta), a_p)$, where $\omega_{p,q}^{\text{fix}}$ denotes the entry in $\omega^{\text{fix}}$ corresponding to the $p$-th description $a_p$ and the $q$-th class $y_q$. $\omega_{p,q}^{\text{fix}}$ is non-zero *only* for $a_p$ that contribute to the aggregation for the class $y_q$. Eq. (2) under this generalized logit definition now becomes:

$$p_{\text{vr}}(y^T|x^T; \delta, \omega^{\text{fix}}, \mathcal{A}) = \frac{\exp([f_{\text{logits}}^{\text{vr}}(x^T; \delta, \omega^{\text{fix}}, \mathcal{A})]_{y^T})}{\sum_{y' \in \mathcal{Y}^T} \exp([f_{\text{logits}}^{\text{vr}}(x^T; \delta, \omega^{\text{fix}}, \mathcal{A})]_{y'})}. \quad (5)$$

**A Decoupling-and-Reweighting Framework.** Considering Eq. (4), a limited $M_a$ in current VR methods may not capture descriptions of diverse aspects from $\mathcal{A}$, shown in Figure 2(a), and a trivial $\omega$ may cause the unreasonable description bias in Figure 2(b).

Therefore, a decoupling-and-reweighting framework like Figure 1(b) can be proposed. Firstly, to enhance learning capacity, a *decoupled* VP set $\Delta = \{\delta_1, \ldots, \delta_v\}$ of size $v$ can be optimized separately with partitions $\mathcal{A}_1, \ldots, \mathcal{A}_v$ of $\mathcal{A}$ to get a better $M_a$. Besides, a more precise Probabilistic Reweighting Matrix $\omega^{\text{PRM}} \in [0, 1]^{|\mathcal{A}| \times |\mathcal{Y}^T|}$ with continuous values can be estimated for *reweighting* the outputs for different attribute descriptions.

## 4. Decoupling Visual Prompts

This section details our DVP method, which implements the decoupling-and-reweighting strategy outlined in Section 3. Figure 3 illustrates the overall DVP pipeline. We begin by describing two ways for partitioning class descriptions (Section 4.1), a preparatory step *before* training. We then explain how our Probabilistic Reweighting Matrix (PRM) is estimated (Section 4.2). We use iterative training, where the VPs set $\Delta$ and the PRM $\omega$ are jointly optimized (see Section 4.3). We lastly show DVP can theoretically attain a lower empirical risk than standard input VR (Section 4.4).

### 4.1. Partitioning Descriptions

A core idea of DVP is to train $v$ distinct VPs $\{\delta_1, \ldots, \delta_v\}$, each specialized for a corresponding partition of the full description set $\{\mathcal{A}_1, \ldots, \mathcal{A}_v\}$. We next present two strategies for partitioning descriptions, namely DVP-cse and DVP-cls.

**By Semantic Causes (DVP-cse).** Following Cai et al. (2025), who used LLMs to generate attribute description sets for each class, we leverage LLMs, e.g., GPT series (Brown et al., 2020), to create semantically coherent description partitions for each cause (i.e., distinguishing aspects for the classification task). First, we query an LLM to identify $v$ primary "causes" (see Appendix A.3). For each $i$-th cause, we formulate a specific query, $\text{cause\_prompt}_i =$"Describe the [$i$-th Cause] of the [Task Info.] image [Class Name]", which is fed into an LLM to generate a set of descriptions $\mathcal{A}_i(y^T)$ for each class $y^T$. The $i$-th global partition is then $\mathcal{A}_i = \bigcup_{y^T \in \mathcal{Y}^T} \mathcal{A}_i(y^T)$. For computational efficiency, these descriptions are pre-converted to text embeddings:

$$\mathcal{E}_i = \{f_{\text{txt}}(a) \mid a \in \mathcal{A}_i\}, \quad \text{for } i = 1, \ldots, v. \quad (6)$$

Each embedding set $\mathcal{E}_i$ (representing $\mathcal{A}_i$) guides the learning of its dedicated prompt $\delta_i$.

**By Unsupervised Clusters (DVP-cls).** Alternatively, when using LLMs to generate descriptions is restricted, we can still partition an *existing* set of descriptions $\mathcal{A}$. We first embed all descriptions into $\mathcal{E} = \{f_{\text{txt}}(a) \mid a \in \mathcal{A}\}$. We then use K-means algorithm (MacQueen et al., 1967) to partition these embeddings into $v$ clusters:

$$\{\mathcal{E}_1, \ldots, \mathcal{E}_v\} = \texttt{KMeans}(\mathcal{E}, v). \tag{7}$$

This way each cluster $\mathcal{E}_i$ implicitly refers to a description partition $\mathcal{A}_i = \{a | f_{\text{txt}}(a) \in \mathcal{E}_i\}$, which is then used to train its respective prompt $\delta_i$.

## 4.2. Reweighting Descriptions

**Probabilistic Reweighting Matrix (PRM).** As discussed in Section 3, PRM accounts for quantifying the relationship between description similarity and class logits. Let $a_p$ be the $p$-the description and $y_q$ be the $q$-th class. We estimate the conditional probability $p(V = a_p | Y^{\text{T}} = y_q)$, denoted by $\omega^{\text{prm}}$. Each entry $\omega_{p,q}^{\text{prm}}$ reflects the contribution of description $a_p$ to class $y_q$. If $a_p$ is not a designated description for class $y_q$ (i.e., $a_p \notin \mathcal{A}(y_q)$), then $\omega_{p,q}^{\text{prm}} = 0$ (Proposition B.1, Appendix B). Otherwise, $\omega_{p,q}^{\text{prm}}$ for $a_p \in \mathcal{A}(y_q)$ can be estimated via MLE from counts on the training set $\mathcal{D}$,

$$\begin{aligned} \omega_{p,q}^{\text{prm}} &= \hat{p}(V = a_p | Y^{\text{T}} = y_q, V \in \mathcal{A}(y_q)) \\ &= \frac{\mathcal{N}_{\mathcal{D}}(a_p, y_q)}{\sum_{a' \in \mathcal{A}(y_q)} \mathcal{N}_{\mathcal{D}}(a', y_q)}. \end{aligned} \tag{8}$$

Here, $\mathcal{N}_{\mathcal{D}}(a, y)$ is the co-occurrence count of description $a$ with class $y$ (see Proposition B.2, Appendix B for details).

**Counting for Few-shot Settings.** To handle data sparsity in few-shot settings (where $\mathcal{N}_{\mathcal{D}}(a_p, y_q)$ might be sparse, making MLE less effective), we smooth these counts. Specifically, for each training sample $(x_j^{\text{T}}, y_j^{\text{T}})$, instead of checking for an exact match with $a_p$, we find the $k$ descriptions in $\mathcal{A}$ most similar to the reprogrammed image $f_{\text{in}}(x_j^{\text{T}} | \Delta)$. Note that only the specific $\delta_i \in \Delta$ is used if a description belongs to $\mathcal{A}_i$. Let this set be $\mathcal{K}(x_j^{\text{T}}, k)$. The count is then:

$$\mathcal{N}_{\mathcal{D}}(a_p, y_q) = \sum_{j=1}^N \mathbf{1}\{y_j^{\text{T}} = y_q\} \cdot \mathbf{1}\{a_p \in \mathcal{K}(x_j^{\text{T}}, k)\}, \tag{9}$$

where $\mathbf{1}\{\cdot\}$ is the indicator function and $k$ is a hyper-parameter. That is, we substitute Eq. (9) into Eq. (8) to obtain a more robust estimation $\omega^{\text{prm}}$ in practice.

## 4.3. Iterative Training

Once the partitioning $\mathcal{E}$ and $\mathcal{A}$ are obtained through either DVP-cse or DVP-cls, the parameters $\Delta$ and $\omega^{\text{prm}}$ are optimized through an alternating optimization procedure (see Algorithm 1 for details). In each training epoch, we first update $\omega^{\text{prm}}$ using Eq. (8) given the current $\Delta$. Then, we update each $\delta_i \in \Delta$ independently using only its assigned

---

**Algorithm 1** Pipeline of DVP

1: **Input:** Few-shot training data $\mathcal{D}^{\text{T}} = \{(x_j^{\text{T}}, y_j^{\text{T}})\}_{j=1}^N$, description set $\mathcal{A}$, hyper-parameters $k, v$, epoch number $E$, and pre-trained CLIP model $f_{\text{clip}}$ with $f_{\text{txt}}$ and $f_{\text{img}}$ encoders
2: **Output:** DVP set $\Delta = \{\delta_1, \ldots, \delta_v\}$ and PRM $\omega^{\text{prm}}$
3:   # Before training: divide descriptions
4: **Obtain** $\mathcal{E} = \bigcup_{i=1}^v \mathcal{E}_i$ by causes (method DVP-cse) using Eq. (6) or clusters (method DVP-cls) using Eq. (7)
5:   # Begin training
6: **Initialize** $\delta_i \leftarrow \mathbf{0}$ for $i = 1, \ldots, v$
7: **for** $e = 1$ **to** $E$ **do**
8:     # Compute/update CLIP output ($M_a$ in Eq. (4))
9:     **Compute** $f_{\text{clip}}(f_{\text{in}}(x_j^{\text{T}} | \Delta), a)$ for $j = 1, \ldots, N, a \in \mathcal{A}$ using $\mathcal{E}$
10:    # Step 1: update reweighting matrix ($\omega$ in Eq. (4))
11:    $\omega \leftarrow \omega^{\text{prm}}$ using Eq. (8) and Eq. (9)
12:    # Compute/update logits output ($M_y$ in Eq. (4))
13:    **Compute** $\tilde{y}_q \leftarrow \sum_{a_p \in \mathcal{A}} \omega_{p,q} f_{\text{clip}}(f_{\text{in}}(x_j^{\text{T}} | \Delta), a_p)$, $q = 1, \ldots, |\mathcal{Y}^{\text{T}}|$ for $j = 1, \ldots, N$
14:    # Step 2: update DVP patterns
15:    $\delta_i \leftarrow \delta_i^*$ using Eq. (10) for $i = 1, \ldots, v$
16: **end for**

---

description partition $\mathcal{A}_i$. Specifically, for each $\delta_i$, the objective mirrors standard VR's objective (Eq. (3)) over $\mathcal{D}$:

$$\min_{\delta_i} \left( -\frac{1}{N} \sum_{j=1}^N \log p_{\text{vr}}(y_j^{\text{T}} | x_j^{\text{T}}; \delta_i, \omega^{\text{prm}}, \mathcal{A}_i) \right). \tag{10}$$

Here, $p_{\text{vr}}$ derives from *partition-specific logits* (using $\mathcal{A}_i$, $\delta_i$ and $\omega^{\text{prm}}$), contrasting a single-prompt aggregation (with $\mathcal{A}$, $\delta$, and $\omega^{\text{fix}}$) as in Eq. (5). This process is applied independently to all $\delta \in \Delta$. See Appendix B.2 for details.

## 4.4. Justification: Empirical Risk Reduction

We analyze DVP by showing its potential to achieve a lower empirical risk on the training data than standard VR.

**Definition 4.1** (Empirical Risk). Consider an input image space $\mathcal{X}$, a discrete label space $\mathcal{Y}$, a distribution $\mathcal{D}'$ defined over $\mathcal{X} \times \mathcal{Y}$. Let $\mathcal{D} = \{(x_j^{\text{T}}, y_j^{\text{T}})\}_{j=1}^N$ be a training set drawn i.i.d from $\mathcal{D}'$. For a classifier $f : \mathcal{X} \to \mathcal{Y}$ parameterized by $\Theta$, the empirical risk $\hat{R}_{\mathcal{D}}(\Theta)$ is defined as the average negative log-likelihood as

$$\hat{R}_{\mathcal{D}}(\Theta) = -\frac{1}{N} \sum_{j=1}^N \log p(y_j | x_j; \Theta).$$

**Empirical Risk of Standard VR.** According to Definition 4.1, for standard VR, the parameters are $\Theta^{\text{vr}} = \{\delta\}$, and the reweighting matrix $\omega^{\text{fix}}$ is pre-determined (e.g., by $\max(\cdot)$ or $\texttt{avg}(\cdot)$ aggregation). Thus, the empirical risk of standard

*Table 1.* Accuracy comparison of different methods trained on 16-shot downstream classification tasks, using ViT-B16-based CLIP as the pretrained model (Mean % ± Std %, ours are  highlighted  and the highest is in **bold**). See Appendix C.7 for parameter numbers.

| METHOD | AIRCRAFT | CALTECH | CARS | DTD | ESAT | FLOWERS | FOOD | PETS | SUN | UCF | RESISC | AVG. |
|--------|----------|---------|------|-----|------|---------|------|------|-----|-----|--------|------|
| VP | $32.1_{\pm 0.6}$ | $93.5_{\pm 0.1}$ | $65.5_{\pm 0.3}$ | $61.4_{\pm 0.5}$ | $91.2_{\pm 0.3}$ | $82.5_{\pm 0.4}$ | $82.3_{\pm 0.1}$ | $91.0_{\pm 0.3}$ | $65.8_{\pm 0.2}$ | $73.8_{\pm 0.5}$ | $79.1_{\pm 0.3}$ | 74.4 |
| AR | $31.7_{\pm 0.3}$ | $95.5_{\pm 0.2}$ | $68.0_{\pm 0.3}$ | $62.0_{\pm 0.1}$ | $93.4_{\pm 0.1}$ | $85.9_{\pm 0.7}$ | $85.2_{\pm 0.1}$ | $92.7_{\pm 0.1}$ | $67.9_{\pm 0.3}$ | $78.1_{\pm 0.2}$ | $81.6_{\pm 0.3}$ | 76.5 |
| ATTRVR | $36.6_{\pm 0.3}$ | $95.7_{\pm 0.1}$ | $68.3_{\pm 0.3}$ | $65.6_{\pm 0.8}$ | $93.8_{\pm 0.3}$ | $92.9_{\pm 0.4}$ | $\mathbf{85.9}_{\pm 0.1}$ | $\mathbf{93.3}_{\pm 0.0}$ | $69.6_{\pm 0.1}$ | $79.0_{\pm 0.6}$ | $82.6_{\pm 0.4}$ | 78.5 |
| DVP-CSE | $\mathbf{40.3}_{\pm 0.2}$ | $\mathbf{96.2}_{\pm 0.1}$ | $\mathbf{72.5}_{\pm 0.2}$ | $\mathbf{66.7}_{\pm 0.4}$ | $93.9_{\pm 0.1}$ | $\mathbf{95.4}_{\pm 0.1}$ | $85.6_{\pm 0.1}$ | $93.1_{\pm 0.0}$ | $\mathbf{71.1}_{\pm 0.2}$ | $81.7_{\pm 0.3}$ | $\mathbf{84.6}_{\pm 0.4}$ | **80.1** |
| DVP-CLS | $38.7_{\pm 0.4}$ | $96.0_{\pm 0.0}$ | $70.8_{\pm 0.2}$ | $65.5_{\pm 0.7}$ | $\mathbf{94.1}_{\pm 0.3}$ | $95.0_{\pm 0.2}$ | $85.7_{\pm 0.0}$ | $\mathbf{93.3}_{\pm 0.1}$ | $\mathbf{71.1}_{\pm 0.2}$ | $\mathbf{82.0}_{\pm 0.1}$ | $84.4_{\pm 0.0}$ | 79.7 |

VR methods is

$$\hat{R}_{\mathcal{D}}^{\mathrm{vr}}(\delta, \omega^{\mathrm{fix}}) = -\frac{1}{N}\sum_{j=1}^{N}\log p_{\mathrm{vr}}(y_j^{\mathrm{T}}|x_j^{\mathrm{T}};\delta,\omega^{\mathrm{fix}},\mathcal{A}),$$

where $p^{\mathrm{vr}}$ is derived from logits in Eq. (1) using $\omega^{\mathrm{fix}}$. $\mathcal{A}$ is the complete set of descriptions.

**Empirical Risk of DVP.** For DVP, whose parameters are $\Theta^{\mathrm{dvp}} = \{\Delta, \omega^{\mathrm{prm}}\}$. Given the risk (Eq. (10)) jointly applied to all $\delta_i \in \Delta$, The empirical risk of DVP is then

$$\hat{R}_{\mathcal{D}}^{\mathrm{dvp}}(\Delta, \omega^{\mathrm{prm}}) = -\frac{1}{N}\sum_{j=1}^{N}\sum_{i=1}^{v}\left[\log p_{\mathrm{vr}}(y_j^{\mathrm{T}}|x_j^{\mathrm{T}};\delta_i,\omega^{\mathrm{prm}},\mathcal{A}_i)\right].$$

*Remark* 4.2. Reducing all $\delta_i \in \Delta$ to the same $\delta$ and substituting $\omega^{\mathrm{prm}}$ with $\omega^{\mathrm{fix}}$ reverts DVP to standard VR.

Given the likelihood-based form of empirical risk (Definition 4.1) used in our context, we further define *optimally achievable empirical risk* to compare the theoretical capability of standard VR and DVP.

**Definition 4.3** (Optimally Achievable Empirical Risk). Consider the training set $\mathcal{D}$ and the classifier $f$ parameterized by $\Theta$ as with Definition 4.1. The optimally achievable empirical risk over $\Theta$ is defined as

$$\hat{R}_{\mathcal{D}}^{*}(\Theta) = \inf_{\Theta}\hat{R}_{\mathcal{D}}(\Theta) = \inf_{\Theta}\left(-\frac{1}{N}\sum_{j=1}^{N}\log p(y_j|x_j;\Theta)\right),$$

where the infimum is taken over all possible parameterizations $\Theta$ of $f$, and, if the infimum is attainable, we denote $\Theta^* = \arg\min_{\Theta}\hat{R}_{\mathcal{D}}(\Theta)$ as the *optimal parameterization* uniquely achieving $\hat{R}_{\mathcal{D}}^{*}(\Theta)$.

We now establish two lemmas and a corollary.

**Lemma 4.4.** *Let $\delta^*$ be the optimal VP for standard VR, minimizing $\hat{R}_{\mathcal{D}}^{\mathrm{vr}}(\delta, \omega^{\mathrm{fix}})$ for a fixed reweighting matrix $\omega^{\mathrm{fix}}$. Let $\Delta_{\omega^{\mathrm{fix}}}^{*}$ be the optimal DVPs that minimizes $\hat{R}_{\mathcal{D}}^{\mathrm{dvp}}(\Delta, \omega^{\mathrm{fix}})$, i.e., DVP using the same fixed $\omega^{\mathrm{fix}}$ as standard VR. Then, we have $\hat{R}_{\mathcal{D}}^{\mathrm{vr}}(\delta^*, \omega^{\mathrm{fix}}) \geq \hat{R}_{\mathcal{D}}^{\mathrm{dvp}}(\Delta_{\omega^{\mathrm{fix}}}^{*}, \omega^{\mathrm{fix}})$.*

Lemma 4.4 (proved in Appendix B.3) demonstrates the advantage of decoupling VPs and training them separately for different description partitions. The decoupling step contributes to the reduction in the optimally achievable empirical risk by improving the learning capacity of the VPs.

**Lemma 4.5.** *For any fixed DVP parameterization $\Delta$, let $\omega^{\mathrm{prm}^*}$ be the optimal PRM that minimizes $\hat{R}_{\mathcal{D}}^{\mathrm{dvp}}(\Delta, \omega^{\mathrm{prm}})$. Then $\hat{R}_{\mathcal{D}}^{\mathrm{dvp}}(\Delta, \omega^{\mathrm{fix}}) \geq \hat{R}_{\mathcal{D}}^{\mathrm{dvp}}(\Delta, \omega^{\mathrm{prm}^*})$ holds for any $\omega^{\mathrm{fix}}$.*

Lemma 4.5 (proved in Appendix B.4) indicates the effectiveness of reweighting the contribution from descriptions to labels. PRM results in a lower empirical risk compared with the pre-defined and fixed weights.

**Corollary 4.6.** *Let $\delta^*$ be the optimal VP for standard VR with $\omega^{\mathrm{fix}}$. Let $\{\Delta^*, \omega^{\mathrm{prm}^*}\}$[1] be the optimal parameterization for DVP that minimizes $\hat{R}_{\mathcal{D}}^{\mathrm{dvp}}(\Delta, \omega^{\mathrm{prm}})$. Then, it holds that $\hat{R}_{\mathcal{D}}^{\mathrm{vr}}(\delta^*, \omega^{\mathrm{fix}}) \geq \hat{R}_{\mathcal{D}}^{\mathrm{dvp}}(\Delta^*, \omega^{\mathrm{prm}^*})$.*

Corollary 4.6 (proved in Append B.5) indicates that the DVP framework, by virtue of its increased capacity through decoupled prompts and adaptive reweighting, has the potential to achieve a lower or at least equal empirical risk on the training data compared to standard VR, effectively preventing underfitting in downstream tasks that may exist when applying VR methods.

## 5. Experiments

**Baselines and Benchmarks.** To evaluate DVP, we follow Cai et al. (2025) to conduct training and testing on pretrained CLIP models (with various image encoder architectures) for 16-shot downstream classification tasks. Averaged accuracies of three seeds are used for evaluation. These tasks cover various classes and domains, including patterns, actions, scenes, etc, with more information in Appendix A.1.

The following three VR baselines are included: (1) VP (Bahng et al., 2022), a VR method in which learnable parameters are overlaid on rescaled downstream images; (2) AR (Tsai et al., 2020; Chen et al., 2023), a VR method that pads learnable parameters around the image; and (3) AttrVR (Cai et al., 2025), a method that guides the learning of VP patterns through attribute descriptions–all baseline methods learn single and non-decoupled VPs.

Both DVP-cse and DVP-cls are tested, with the hyperparameter $k$ chosen to be 3 (see Table 7). The number of VPs in DVP-cse is set to be $v = 3$, while that in DVP-cls is a

---

[1]We note that $\Delta^*$ is different from $\Delta_{\omega^{\mathrm{fix}}}^{*}$ in Lemma 4.4.

*Table 2.* Ablation studies of DVP-cse and DVP-cls, using ViT-B16-based CLIP as the pretrained model (Mean % ± Std %, ours are highlighted and the highest is in **bold**).

| Method | Aircraft | Caltech | Cars | DTD | ESAT | Flowers | Food | Pets | SUN | UCF | Resisc | Avg. |
|---|---|---|---|---|---|---|---|---|---|---|---|---|
| DVP-CSE | **40.3**$_{\pm0.2}$ | 96.2$_{\pm0.1}$ | 72.5$_{\pm0.2}$ | 66.7$_{\pm0.4}$ | 93.9$_{\pm0.1}$ | **95.4**$_{\pm0.1}$ | 85.6$_{\pm0.1}$ | 93.1$_{\pm0.0}$ | 71.1$_{\pm0.2}$ | **81.7**$_{\pm0.3}$ | 84.6$_{\pm0.4}$ | **80.1** |
| W/O CSE | 36.4$_{\pm0.2}$ | 95.8$_{\pm0.1}$ | 69.1$_{\pm0.1}$ | 65.3$_{\pm0.7}$ | 94.1$_{\pm0.3}$ | 93.6$_{\pm0.2}$ | 85.7$_{\pm0.1}$ | 93.1$_{\pm0.0}$ | 70.0$_{\pm0.1}$ | 80.2$_{\pm0.3}$ | 82.8$_{\pm0.4}$ | 78.7 |
| W/O VR | 27.5$_{\pm0.3}$ | 93.2$_{\pm0.3}$ | 63.9$_{\pm0.4}$ | 57.0$_{\pm0.3}$ | 45.9$_{\pm0.8}$ | 83.6$_{\pm0.2}$ | 83.9$_{\pm0.0}$ | 91.3$_{\pm0.1}$ | 62.3$_{\pm0.2}$ | 68.3$_{\pm0.3}$ | 60.3$_{\pm0.4}$ | 67.0 |
| W/O $\omega^{\mathrm{prm}}$ | 38.0$_{\pm0.1}$ | 95.8$_{\pm0.1}$ | 70.4$_{\pm0.4}$ | **67.7**$_{\pm0.3}$ | 94.8$_{\pm0.1}$ | 95.0$_{\pm0.3}$ | 85.8$_{\pm0.1}$ | 93.5$_{\pm0.1}$ | 67.1$_{\pm0.1}$ | 80.3$_{\pm0.5}$ | **84.9**$_{\pm0.1}$ | 79.4 |
| DVP-CLS | 38.7$_{\pm0.4}$ | 96.0$_{\pm0.0}$ | 70.8$_{\pm0.2}$ | 65.5$_{\pm0.7}$ | 94.1$_{\pm0.3}$ | 95.0$_{\pm0.2}$ | 85.7$_{\pm0.0}$ | 93.3$_{\pm0.1}$ | 71.1$_{\pm0.2}$ | 82.0$_{\pm0.1}$ | 84.4$_{\pm0.0}$ | **79.7** |
| W/O CLS | 36.4$_{\pm0.2}$ | 95.8$_{\pm0.1}$ | 69.1$_{\pm0.1}$ | 65.3$_{\pm0.7}$ | 94.1$_{\pm0.3}$ | 93.6$_{\pm0.2}$ | 85.7$_{\pm0.1}$ | 93.1$_{\pm0.0}$ | 70.0$_{\pm0.1}$ | 80.2$_{\pm0.3}$ | 82.8$_{\pm0.4}$ | 78.7 |
| W/O VR | 26.0$_{\pm0.0}$ | 93.9$_{\pm0.1}$ | 63.1$_{\pm0.2}$ | 55.0$_{\pm0.2}$ | 52.2$_{\pm0.4}$ | 83.1$_{\pm0.3}$ | 84.8$_{\pm0.0}$ | 91.3$_{\pm0.1}$ | 64.4$_{\pm0.1}$ | 70.0$_{\pm0.4}$ | 60.8$_{\pm0.5}$ | 67.7 |
| W/O $\omega^{\mathrm{prm}}$ | 37.8$_{\pm0.4}$ | 96.0$_{\pm0.2}$ | 70.6$_{\pm0.1}$ | 65.4$_{\pm0.7}$ | 93.9$_{\pm0.2}$ | 94.2$_{\pm0.3}$ | 85.8$_{\pm0.1}$ | 93.3$_{\pm0.1}$ | 69.4$_{\pm0.1}$ | 80.7$_{\pm0.2}$ | **84.6**$_{\pm0.1}$ | 79.2 |

*Figure 4.* Accuracy comparison of different VR methods with the number of VR patterns (i.e., VPs) $v \in \{1, 2, 3, 5, 7\}$. Pre-trained ViT-B16-based CLIP is used. The striped area indicates the error bars.

hyper-parameter that represents unsupervised clusters and is chosen as shown in Table 8. More information about hyper-parameters is in Appendix C.1. Implementation details are in Appendix A.2 and generated causes are in Appendix A.3.

**Performance Comparison.** The results on the ViT-16-based CLIP model are shown in Table 1. It can be observed that both DVP-cse and DVP-cls demonstrate strong performance. DVP-cse outperforms the baseline method AttrVR by an average of 1.6% across 11 datasets, while DVP-cls achieves an average improvement of 1.2%. Except for the Food dataset, applying DVP-cse or DVP-cls achieves accuracy equal to or higher than AttrVR on all other datasets.

DVP-cse is designed to train VPs for descriptions grouped by different causes. Consequently, it achieves better results when multiple specific aspects are crucial for classification. For instance, the Cars, Aircraft, and Flowers datasets contain fine-grained classes that vary along multiple cause factors (i.e., color, shape), allowing DVP-cse to achieve improvements of 4.2%, 3.7%, and 2.5% over AttrVR, respectively.

Meanwhile, DVP-cls is a method that learns VPs based on descriptions after unsupervised clustering. It tends to group classes with similar features into a cluster for one single VP. For datasets like UCF (i.e., different actions), or fine-grained datasets like Cars and Aircraft, classes can be clearly divided into subcategories based on similarity. Therefore, DVP-cls achieves better performance than AttrVR, by 3%, 2.5%, and 2.1% on these datasets, respectively.

DVP shows no improvement on the Food dataset. This might be attributed to the primary distinguishing factors for food being smell and taste, while visual information is less useful. As a result, decoupling VPs for better visual learning will not lead to classification improvements.

**Ablation Studies.** The ablation studies for DVP-cse and DVP-cls are presented in Table 2. "W/o cse" and "w/o cls" refer to excluding decoupled VPs, reducing both methods to training a single VP. "W/o VR" indicates the zero-shot learning results without training the VR pattern, while "w/o $\omega^{\mathrm{prm}}$" represents results without PRM.

Without decoupling, a single VP struggles to capture diverse descriptions, leading to unsatisfactory performance, especially on complex fine-grained classification tasks such as Aircraft, Cars, and Flowers. Similarly, omitting the optimization of VP results in poor accuracy on downstream tasks with significant differences from the pre-training domain, such as remote sensing datasets (e.g., EuroSAT, Resisc). When the reweighting matrix $\omega^{\mathrm{prm}}$ is not applied, the performance is notably affected in datasets with a large number of categories (e.g., SUN with 397 classes) or complex descriptions (e.g., UCF for action classification). In summary, every component of DVP is indispensable for achieving optimal performance improvements.

**Impact of the Number of VPs.** Figure 4 illustrates the results of VR methods with various numbers of VPs. Simply increasing the number of VPs without employing the

Weight Proportions of Different Causes after Training VR Patterns for Some Classes (Downstream Task: Flowers ; Method: DVP-cse)

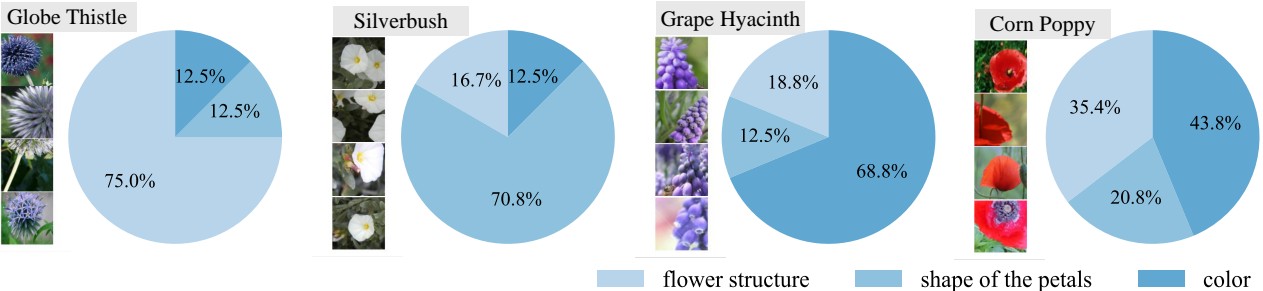

flower structure        shape of the petals        color

*Figure 5.* Visualization results of DVP-cse, showing weights of causes for classifying certain classes, using ViT-B16-based CLIP.

Top Classes within Different Clusters (Downstream Task: Flowers; Method: DVP-cls)

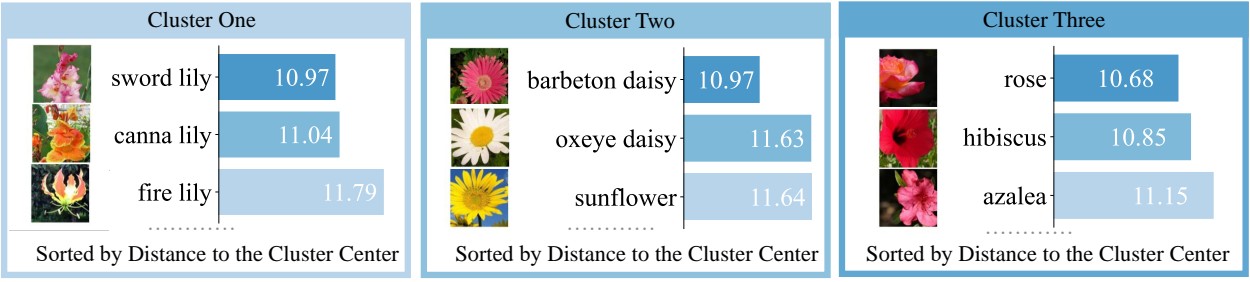

*Figure 6.* Visualization results of DVP-cls, showing the top 3 classes closest to the description cluster centers, using ViT-B16-based CLIP.

decoupling-and-reweighting framework proposed in this paper may even lead to a decrease in accuracy due to potential overfitting. However, when the DVP approach is used–whether in the form of DVP-cse or DVP-cls–the accuracy steadily improves. This suggests that the performance improvement achieved by DVP is not merely due to the increase in parameters, but is attributable to the benefits of our learning framework.

**Visualization of Cause Weights (DVP-cse).** In DVP-cse, since each trained VP corresponds to a specific cause, for a single class, the weights corresponding to individual VPs in the reweight matrix can be summed to calculate the contribution of that cause to the predicted label. Figure 5 illustrates the contributions of three causes for four classes in the Flowers dataset. Specifically, the spherical spiny shape of the globe thistle, the trumpet-shaped petals of silverbush, and the bluish-purple petals of grape hyacinth make these three flowers distinct, matching the statistical results–flower structure, petal shape, and color are the primary causes for classifying these flowers, respectively. In contrast, corn poppy is relatively ordinary in various aspects, resulting in similar contributions across different reasons. It can be observed that the reprogramming process determines the priority of causes driving the final class predictions. More results on Texture and Aircraft datasets are detailed in Appendix C.2.

*Table 3.* Average accuracy of different VR methods on 11 datasets, using different backbones as CLIP visual encoders (Mean Accuracy %, ours are highlighted and the highest is in **bold**, RN stands for ResNet).

|  | VP | AR | ATTRVR | DVP-CSE | DVP-CLS |
|---|---|---|---|---|---|
| RN50 | 53.5 | 60.4 | 64.6 | **66.6** | 66.0 |
| RN101 | 57.5 | 62.7 | 67.2 | **69.6** | 68.8 |
| ViT-B32 | 68.3 | 66.3 | 69.8 | **71.3** | 71.0 |
| ViT-B16 | 74.4 | 76.5 | 78.5 | **80.1** | 79.7 |

**Visualization of Cluster Components (DVP-cls).** Compared with the VPs in DVP-cse, which have clear correspondences to specific causes, the VPs in DVP-cls tend to correspond to descriptions of classes with commonalities. Figure 6 shows the classes with the closest average distance to the description cluster centers corresponding to different VPs. It can be observed that different types of lilies are grouped into one cluster, while daisies and sunflowers with similar shapes are grouped together, and flowers with similar colors are also grouped together. This explains the results in Table 1, where DVP-cse is more suitable for classification tasks with multiple contributing causes, while DVP-cls is better suited for classification tasks with distinct subclass hierarchies. More results are detailed in Appendix C.3.

**Results on Different Backbones.** Since VR methods modify only the input images, they are compatible with any

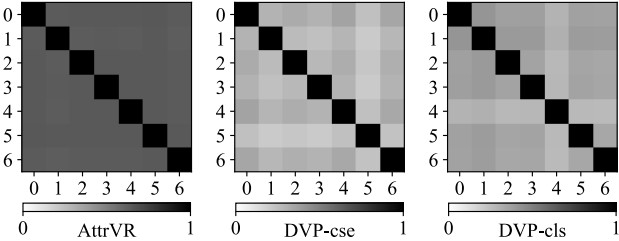

*Figure 7.* Pairwise HSIC between sets of image embeddings applying non-decoupled (i.e., AttrVR) or decoupled VPs (i.e., our DVP). Experiments are done on the Texture task with ViT-16-based CLIP.

model architecture. Table 3 presents the classification accuracy averaged across all datasets when using different image encoders (i.e., ResNet or ViT) in CLIP (detailed results are in Appendix C.4). It can be observed that both DVP-cse and DVP-cls achieve improvements over all baseline methods, regardless of whether smaller (RN50) or larger (ViT-B16) pre-trained models are used. The performance improvement of DVP becomes more pronounced for smaller image encoders, indicating that DVP can compensate for the limitations of pretrained models.

**Discussion about Independence.** We further investigate the independence of different VPs using the *Hilbert-Schmidt Independence Criterion* (HSIC) (Gretton et al., 2007). HSIC is a non-parametric test where smaller values indicate greater statistical independence between two random variables. We calculate pairwise HSIC values (with $\gamma = 3$) between the embedding distributions of images after adding different VPs. This is done for a baseline approach (i.e., AttrVR, which does not decouple VPs) and the proposed decoupling-based approaches (i.e., our DVP implemented under both DVP-cse and DVP-cls). Experiments are conducted on the Texture task, which features relatively simple images to highlight the differences. Results in Figure 7 show that embeddings obtained by decoupled VPs exhibit lower statistical dependence between different VPs, which might be a factor that contributes to the improved performance.

**More Experiments.** We conduct several additional experiments to further validate our approach and explore its characteristics. In Appendix C.5, we analyze the time cost of various VR methods and conclude that DVP can achieve performance improvements under the same time cost constraints. In Appendix C.6, we analyze different causes and investigate what makes a cause effective for VR using DVP-cse. We conclude that the most suitable causes should (1) be related to visual information, and (2) have a causal relationship with accurate classification. This experiment demonstrates the application of DVP-cse in evaluating the importance of various causes for VR in classification tasks. In Appendix C.7, we discuss the potential parameter overhead associated with DVP, and introduce a vari-

ant—DVPlite—which decouples VPs without introducing additional parameters, confirming that our decoupling-and-reweighting strategy improves performance even when no extra parameters are used. In Appendix C.8, we demonstrate the applicability of our DVP-PRM framework to different VR methods. In Appendix C.9, we lastly present results showing how performance varies with different amounts of available training data.

## 6. Conclusion

In this paper, to better align downstream images with descriptions for classification, we proposed a decoupling-and-reweighting framework that learns DVP based on specific causes or unsupervised clusters, combined with a PRM to reweight the descriptions. DVP is shown to be effective both theoretically and empirically. By exploring and analyzing the distinct roles and contributions of individual VPs, we offer new insights into model reprogramming.

## Acknowledgement

CYC, ZSY, and FL are supported by the Australian Research Council (ARC) with grant number DE240101089, and FL is also supported by ARC with grant number LP240100101, DP230101540 and the NSF&CSIRO Responsible AI program with grant number 2303037. Jianzhong Qi is supported in part by the ARC via Discovery Project DP240101006 and Future Fellowship FT240100170. This research is also supported by The University of Melbourne's Research Computing Services and the Petascale Campus Initiative. We sincerely appreciate the time and dedication of the reviewers in carefully reviewing our manuscript.

## Impact Statement

This paper presents work whose goal is to advance the field of Machine Learning. There are many potential societal consequences of our work, none of which we feel must be specifically highlighted here.

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

# A. Appendix 1: More Training Information

## A.1. Dataset Information

Table 4. Dataset Information

| | AIRCRAFT | CALTECH | CARS | DTD | ESAT | FLOWERS | FOOD | PETS | SUN | UCF | RESISC |
|---|---|---|---|---|---|---|---|---|---|---|---|
| TASK INFO. | aircraft model | object | fine-grained automobile | texture | remote sensing land cover | flower | food | pet | scene | action | remote sensing scene |
| CLASS NUM. | 100 | 100 | 196 | 47 | 10 | 102 | 101 | 37 | 397 | 101 | 45 |
| BATCH SIZE | 64 | 64 | 64 | 64 | 64 | 64 | 64 | 64 | 64 | 64 | 64 |

We follow the prior work (Cai et al., 2025) to set up our benchmark, employing the same methodology to split the 16-shot training, validation, and test sets. All datasets are publicly available and listed as follows: FGVCAircraft (Aircraft) (Maji et al., 2013), Caltech101 (Caltech) (Fei-Fei et al., 2004), StanfordCars (Cars) (Krause et al., 2013), Texture (DTD) (Cimpoi et al., 2014), EuroSAT (ESAT) (Helber et al., 2019), Flowers102 (Flowers) (Nilsback & Zisserman, 2008), Food101 (Food) (Bossard et al., 2014), OxfordPets (Pets) (Parkhi et al., 2012), SUN397 (SUN) (Xiao et al., 2010), UCF101 (UCF) (Soomro et al., 2012), Resisc45 (Resisc) (Cheng et al., 2017). Detailed task information and the batch size used for training VR are provided in Table 4.

## A.2. Implement Details

All VR baseline methods are trained with consistent settings: a learning rate of 40, a momentum of 0.9 (SGD optimizer (Harold et al., 1997)), and a cosine annealing scheduler (Loshchilov & Hutter, 2016), over 200 epochs. Results are averaged across three random seeds. For method-specific hyper-parameters, we followed (Cai et al., 2025) by using a VR noise pattern with a frame size of 30 for VP (Bahng et al., 2022) and a frame size of 16 for AR (Chen et al., 2023; Tsai et al., 2020) and AttrVR (Cai et al., 2025). To ensure fairness, our DVP utilized the same settings as AttrVR.

For DVP-cls, we use the same descriptions as (Cai et al., 2025). For K-means, we use a maximum iteration of 300 and the relative tolerance regarding the Frobenius norm of differences in cluster centers to be 1e-4. For DVP-cse, we use GPT-4o-mini (Brown et al., 2020) to generate descriptions, with the maximum token to be 50, stopped at '.', and the temperature to be 0.99. We set $m = 20$ for each cause number in DVP-cse and the attribute numbers in DVP-cls.

## A.3. Generated Causes

Table 5. Top Three Causes Used by DVP-cse for Each Dataset

| | CAUSE 1 | CAUSE 2 | CAUSE 3 |
|---|---|---|---|
| AIRCRAFT | shape and structure | size and proportions | surface features |
| CALTECH | shape | color | texture |
| CARS | front and rear design | shape and body style | wheels and rims |
| DTD | pattern | coarseness/granularity | contrast |
| ESAT | texture | spectral information | spatial characteristics |
| FLOWERS | flower structure | shape of the petals | color |
| FOOD | texture | shape and composition | color |
| PETS | fur or coat type | species and body shape | facial features |
| SUN | spatial layout and composition | texture | color and lighting |
| UCF | motion and poses | contextual elements | intensity and speed |
| RESISC | spectral information | texture | spatial characteristics |

For DVP-cse, the top three causes for each dataset are obtained by asking an LLM to 'List the top three causes for classifying [Task Info.] images', with the specific results presented in Table 5. Notably, to investigate the impact of different causes on the results, we generated the top seven causes for the Aircraft, DTD, and Flowers datasets for comparison, as listed in Table 6. The results of DVP-cse in Figure 4 and Figure 7 apply these causes. The effect of different causes will be discussed in Appendix C.6.

*Table 6.* Top Seven Causes Used by for Aircraft, DTD and Flowers Dataset

|  | AIRCRAFT | DTD | FLOWERS |
|---|---|---|---|
| CAUSE 1 | shape and structure | pattern | flower structure |
| CAUSE 2 | size and proportions | coarseness/granularity | shape of the petals |
| CAUSE 3 | surface features | contrast | color |
| CAUSE 4 | engine configuration | directionality | size of the Flower |
| CAUSE 5 | wing configuration | regularity | fragrance |
| CAUSE 6 | tail design | roughness | leaf characteristics |
| CAUSE 7 | landing gear configuration | entropy | flowering time and growth habit |

## B. Appendix 2: More Theoretical Justification

### B.1. Detailed Propositions and Proof

**Proposition B.1.** *Assuming that $V$ is variable representing the description text and $Y^{\mathrm{T}}$ is the variable representing the downstream label, where $V \in \mathcal{A}, Y^{\mathrm{T}} \in \mathcal{Y}^{\mathrm{T}}$, and $\mathcal{A}(Y^{\mathrm{T}}) \subseteq \mathcal{A}$ is the description subset that corresponds to class $Y^{\mathrm{T}}$, it can be obtained that $\forall V \notin \mathcal{A}(Y^{\mathrm{T}})$, $p(V|Y^{\mathrm{T}}) = 0$ and $\forall V \in \mathcal{A}(Y^{\mathrm{T}})$, $p(V|Y^{\mathrm{T}}) = p(V|Y^{\mathrm{T}}, V \in \mathcal{A}(Y^{\mathrm{T}}))$.*

*Proof.* By the law of total probability (Kolmogorov, 1956), we know

$$
\begin{aligned}
p(V|Y^{\mathrm{T}}) =& p(V|Y^{\mathrm{T}}, V \in \mathcal{A}(Y^{\mathrm{T}})) \cdot p(V \in \mathcal{A}(Y^{\mathrm{T}})|Y^{\mathrm{T}}) \\
&+ p(V|Y^{\mathrm{T}}, V \notin \mathcal{A}(Y^{\mathrm{T}})) \cdot p(V \notin \mathcal{A}(Y^{\mathrm{T}})|Y^{\mathrm{T}}).
\end{aligned}
\tag{11}
$$

For all attribute descriptions that not correspond to label $y^{\mathrm{T}}$, (i.e, $\forall V \notin \mathcal{A}(y^{\mathrm{T}})$), we have $p(V|Y^{\mathrm{T}}) = 0$, as $V$ is the description irrelevant to $Y^{\mathrm{T}}$.

Besides, for attribute descriptions that correspond to $y^{\mathrm{T}}$, we have

$$
\begin{aligned}
\forall V \in \mathcal{A}(y^{\mathrm{T}}), p(V \in \mathcal{A}(Y^{\mathrm{T}})|Y^{\mathrm{T}}) = 1, \\
\forall V \in \mathcal{A}(y^{\mathrm{T}}), p(V \notin \mathcal{A}(Y^{\mathrm{T}})|Y^{\mathrm{T}}) = 0,
\end{aligned}
\tag{12}
$$

By substituting the above two equation into Eq. (11), it can be obtained that $\forall V \in \mathcal{A}(y^{\mathrm{T}})$,

$$
\begin{aligned}
p(V|Y^{\mathrm{T}}) =\ & p(V|Y^{\mathrm{T}}, V \in \mathcal{A}(Y^{\mathrm{T}})) \times 1 \\
&+ p(V|Y^{\mathrm{T}}, V \notin \mathcal{A}(Y^{\mathrm{T}})) \times 0 \\
=\ & p(V|Y^{\mathrm{T}}, V \in \mathcal{A}(Y^{\mathrm{T}})).
\end{aligned}
$$

Conclusively, we get:

$$
p(V|Y^{\mathrm{T}}) = \begin{cases} p(V|Y^{\mathrm{T}}, V \in \mathcal{A}(Y^{\mathrm{T}})) & V \in \mathcal{A}(Y^{\mathrm{T}}) \\ 0 & V \notin \mathcal{A}(Y^{\mathrm{T}}) \end{cases}
$$

as is stated in Proposition B.1. $\square$

**Proposition B.2.** *Assuming $N(\cdot, \cdot)$ is the counting function that returns the frequency of two items occurring together, $V \in \mathcal{A}, Y^{\mathrm{T}} \in \mathcal{Y}^{\mathrm{T}}$ are description and label variables respectively, $\mathcal{A}(Y^{\mathrm{T}}) \subseteq \mathcal{A}$ is the description subset that corresponds to class $Y^{\mathrm{T}}$ and $a, y^{\mathrm{T}}$ are values, then through MLE (Fisher, 1922), the conditional probability $p(V = a|Y^{\mathrm{T}} = y^{\mathrm{T}}, V \in \mathcal{A}(y^{\mathrm{T}}))$ can be estimated on training set $\mathcal{D}$ as*

$$
\hat{p}(V = a|Y^{\mathrm{T}} = y^{\mathrm{T}}, V \in \mathcal{A}(y^{\mathrm{T}})) = \frac{\mathcal{N}_{\mathcal{D}}(V = a, Y^{\mathrm{T}} = y^{\mathrm{T}})}{\sum_{a' \in \mathcal{A}(y^{\mathrm{T}})} \mathcal{N}_{\mathcal{D}}(V = a', Y^{\mathrm{T}} = y^{\mathrm{T}})}.
$$

*Proof.* According to the definition of conditional probability, we can rewrite $p(V = a|Y^{\mathrm{T}} = y^{\mathrm{T}}, a \in \mathcal{A}(y^{\mathrm{T}}))$ as:

$$
\begin{aligned}
p(V = a|Y^{\mathrm{T}} = y^{\mathrm{T}}, V \in \mathcal{A}(y^{\mathrm{T}})) &= \frac{p(V = a, Y^{\mathrm{T}} = y^{\mathrm{T}}, V \in \mathcal{A}(y^{\mathrm{T}}))}{p(Y^{\mathrm{T}} = y^{\mathrm{T}}, V \in \mathcal{A}(y^{\mathrm{T}}))} \\
&= \frac{p(V = a, a \in \mathcal{A}(y^{\mathrm{T}})|Y^{\mathrm{T}} = y^{\mathrm{T}})}{p(V \in \mathcal{A}(y^{\mathrm{T}})|Y^{\mathrm{T}} = y^{\mathrm{T}})}.
\end{aligned}
$$

Therefore, for $\forall a \in \mathcal{A}(y^{\mathrm{T}})$:

$$
p(V = a|Y^{\mathrm{T}} = y^{\mathrm{T}}, V \in \mathcal{A}(y^{\mathrm{T}})) = \frac{p(V = a|Y^{\mathrm{T}} = y^{\mathrm{T}})}{\sum_{a' \in \mathcal{A}(y^{\mathrm{T}})} p(V = a'|Y^{\mathrm{T}} = y^{\mathrm{T}})}, \tag{13}
$$

where $p(V = a|Y^{\mathrm{T}} = y^{\mathrm{T}})$ is a value in $\omega$ to be estimated. Given the downstream training set $\mathcal{D} = \{(x_i^{\mathrm{T}}, y_i^{\mathrm{T}})\}_{i=1}^{N}$, the VR method and description set $\mathcal{A}$, the maximum likelihood $\mathcal{L}(\omega)$ is formulated to be

$$
\mathcal{L}(\omega) = \prod_{i=1}^{N} p(V = a_i|Y^{\mathrm{T}} = y_i^{\mathrm{T}}), \tag{14}
$$

where $a_i$ is the most similar description for sample $x_i^{\mathrm{T}}$, given the input DVP pattern $\Delta$ and description set $\mathcal{A}$. For simplicity, we use $w_{p,q} \triangleq p(a_p|y_q)$ to represent $p(V = a_p|Y^{\mathrm{T}} = y_q)$. Thus, according to Eq. (14) any single item $\omega_{p,q}$ in $\omega$ will have the maximum likelihood as

$$
\mathcal{L}(\omega_{p,q}) = \prod_{Y^{\mathrm{T}} \in \mathcal{Y}^{\mathrm{T}}} \prod_{V \in \mathcal{A}} p(V = a_p|, Y^{\mathrm{T}} = y_q)^{\mathcal{N}_{\mathcal{D}}(a_p, y_q)},
$$

where $\mathcal{N}_{\mathcal{D}}(a_p, y_q)$ is the counting function that returns the frequency of matched $a_p$ and $y_q$ pairs. Taking the logarithm gives:

$$
l(\omega_{p,q}) = \sum_{Y^{\mathrm{T}} \in \mathcal{Y}^{\mathrm{T}}} \sum_{V \in \mathcal{A}} \mathcal{N}_{\mathcal{D}}(a_p, y_q) \cdot \log p(V = a_p|, Y^{\mathrm{T}} = y_q), s.t. \sum_{a' \in \mathcal{A}} p(V = a'|, Y^{\mathrm{T}} = y_q) = 1.
$$

Then we introduce Lagrange multipliers $\lambda_q$ for constrained optimization:

$$
l'(\omega_{p,q}, \lambda) = l(\omega_{p,q}) + \lambda_q(1 - \sum_{a' \in \mathcal{A}} p(V = a'|Y^{\mathrm{T}} = y_q)).
$$

Taking partial derivatives on both sides of the equation, we get

$$
\frac{\partial l'}{\partial p(V = a_p|Y^{\mathrm{T}} = y_q)} = \frac{\mathcal{N}_{\mathcal{D}}(a_p, y_q)}{p(V = a_p|Y^{\mathrm{T}} = y_q)} - \lambda_q = 0.
$$

Therefore, $\hat{p}(V = a_p|, Y^{\mathrm{T}} = y_q) = \frac{\mathcal{N}_{\mathcal{D}}(a_p, y_q)}{\lambda_q}$. Since $\sum_{a_p \in \mathcal{A}} p(V = a_p|, Y^{\mathrm{T}} = y_q) = 1$, we get $\sum_{a_p \in \mathcal{A}} \frac{\mathcal{N}_{\mathcal{D}}(a_p, y_q)}{\lambda_q} = 1$ and thus $\lambda_q = \mathcal{N}_{\mathcal{D}}(y_q)$. Substituting back into Eq. (13), we get the final prediction:

$$
\hat{p}(V = a_p|Y^{\mathrm{T}} = y_q) = \frac{\mathcal{N}_{\mathcal{D}}(a_p, y_q)}{\sum_{a' \in \mathcal{A}(y_q)} \mathcal{N}_{\mathcal{D}}(a', y_q)}.
$$

$\square$

## B.2. Relationship between $p_{\mathrm{vr}}$ and $p_{\mathrm{dvp}}$

In DVP, since $\mathcal{A} = \bigcup_{i=1}^{v}$ and $\mathcal{A}(y_i^{\mathrm{T}}) \cap \mathcal{A}(y_j^{\mathrm{T}}) = \varnothing$ for $i \neq j$, the total similarity between $x^{\mathrm{T}}$ and $\mathcal{A}$ decomposes additively across partitions. Moreover, each $\delta_i$ optimizes only for $\mathcal{A}_i$, making $\delta_i$ exclusive prompt responsible for aligning $x^{\mathrm{T}}$ with descriptions in $\mathcal{A}_i$.

The integrated logits of all prompts $\Delta$ can thus be expressed as a summation over all individual $\delta_i$ with respect to the designated description partition $\mathcal{A}_i$.

$$\underbrace{\left[ f_{\text{logits}}^{\text{dvp}}(x^{\text{T}}; \Delta, \omega^{\text{prm}}) \right]_{y_q^{\text{T}}}}_{\text{integrated logit}} = \sum_{i=1}^{v} \left[ f_{\text{logits}}^{\delta_i}(x^{\text{T}}; \delta_i, \omega^{\text{prm}}) \right]_{y_q^{\text{T}}}$$

$$= \sum_{i=1}^{v} \underbrace{\sum_{a_p \in \mathcal{A}_i} \left( f_{\text{clip}}(f_{\text{in}}(x^{\text{T}}|\delta_i), a_p) \right)_{y_q^{\text{T}}} \cdot \omega_{p,q}^{\text{prm}}}_{\text{partition-specific logit contribution}}.$$

Given the form of Eq. (B.2), it is easy to find that, by reducing $\Delta = \{\delta_1, \ldots, \delta_v\}$ to the same $\delta$, as well as substituting $\omega^{\text{prm}}$ with $\omega^{\text{fix}}$ (i.e., $\text{agg}(\cdot)$), DVP reverts back to standard VR.

### B.3. Proof of Lemma 4.4

*Proof.* The standard VR method, parameterized by a single prompt $\delta$, can be viewed as a special case of the DVP model (both using a fixed $\omega^{\text{fix}}$) where all decoupled prompts are constrained to be identical, i.e., $\delta_1 = \delta_2 = \ldots, \delta_v = \delta$. Specifically, let $\Delta^{\text{tied}} = \{\delta, \ldots, \delta\}$, then the DVP logits using $\omega^{\text{fix}}$ become:

$$\left[ f_{\text{logits}}^{\text{dvp}}(x^{\text{T}}; \Delta^{\text{tied}}, \omega^{\text{fix}}) \right]_{Y^{\text{T}}=y_q^{\text{T}}} = \sum_{i=1}^{v} \sum_{a_p \in \mathcal{A}_s} \left( f_{\text{clip}}(f_{\text{in}}(x^{\text{T}}|\delta), a_p) \cdot \omega_{p,q}^{\text{fix}} \right).$$

Since the partitions $\mathcal{A}_i$ are *disjoint* and their union is $\mathcal{A}$, this sum is equivalent to $\sum_{a_p \in \mathcal{A}} \left( f_{\text{clip}}(f_{\text{in}}(x^{\text{T}}|\delta), a_p) \cdot \omega_{p,q}^{\text{fix}} \right)$. This expression, when aggregated according to the rules defined in $\omega^{\text{fix}}$, e.g., $\max(\cdot)$ or $\text{avg}(\cdot)$ over descriptions for class $y_q^{\text{T}}$, yields the logits of the standard VR model $f_{\text{logits}}^{\text{dvp}}(x^{\text{T}}; \delta, \mathcal{A})$. Therefore, the set of functions representable by standard VR (parameterized by $\delta$) is a subset of those representable by DVP (parameterized by $\Delta$) when both use the same $\omega^{\text{fix}}$.

Let $\mathcal{F}_{\text{vr}}$ be the function space spanned by standard VR prompts and $\mathcal{F}_{\text{dvp}}$ be that for DVP prompts (both with fixed $\omega^{\text{fix}}$), then $\mathcal{F}_{\text{vr}} \subseteq \mathcal{F}_{\text{dvp}}$. Given that $\delta^*$ and $\Delta_{\omega^{\text{fix}}}^*$ represent the *optimally achievable parameterization* within $\mathcal{F}_{\text{vr}}$ and $\mathcal{F}_{\text{dvp}}$, respectively, it is ensured that $\hat{R}_{\mathcal{D}}^{\text{vr}}(\delta^*, \omega^{\text{fix}}) \geq \hat{R}_{\mathcal{D}}^{\text{dvp}}(\Delta_{\omega^{\text{fix}}}^*, \omega^{\text{fix}})$ because $\min_{f \in \mathcal{F}_A} R_{\mathcal{D}}^*(f) \leq \min_{f' \in \mathcal{F}_B} R_{\mathcal{D}}^*(f')$, s.t. $\mathcal{F}_B \subseteq \mathcal{F}_A$ (based on Definition 4.3 and the set inclusion relationship). This complete the proof.

$\square$

### B.4. Proof of Lemma 4.5

*Proof.* Assuming the hypothesis space of $\omega^{\text{prm}}$ to be $\Phi^{\text{prm}}$, and the optimized $\omega^{\text{prm}} \in \Phi^{\text{prm}}$ to be $\omega^{\text{prm}*}$, then for a fixed $\Delta$, we have:

$$\forall \omega \in \Phi^{\text{prm}} : \hat{R}_{\mathcal{D}}^{\text{dvp}}(\Delta, \omega) \geq \hat{R}_{\mathcal{D}}^{\text{dvp}}(\Delta, \omega^{\text{prm}*}). \tag{15}$$

We define the $\omega$ in standard VR method that use $\max(\cdot)$ in Eq. (1) to be $\omega^{\text{max}}$, where $\omega^{\text{max}} \in \{0, 1\}^{|\mathcal{A}| \times |\mathcal{Y}^{\text{T}}|}$ as discussed in Section 3. $\omega_{p,q}^{\text{max}}$ is set to be 1 only when $a_p$ is the $q$-th class's attribute description with the maximum CLIP output, otherwise $\omega_{p,q}^{\text{max}} = 0$.

Besides, when $\text{avg}(\cdot)$ is used in Eq. (1), we define $\omega = \omega^{\text{avg}}$. Then $\omega^{\text{avg}} \in \{0, \frac{1}{m}\}^{|\mathcal{A}| \times |\mathcal{Y}^{\text{T}}|}$, where $\omega_{p,q}^{\text{avg}} = \frac{1}{m}$ only when the $p$-th attribute is the description of the $q$-th class. As a result, $\omega^{\text{fix}} \in \{\omega^{\text{max}}, \omega^{\text{avg}}\}$ for standard VR methods.

For $\omega^{\text{max}}$, we have:

$$\omega_{p,q}^{\text{max}} = \hat{p}(V = a_p | Y^{\text{T}} = y_q) = \begin{cases} 1 & a_p = \arg\max_{a' \in \mathcal{A}(y_q)} \mathbb{E}_{\mathcal{D}}(\tilde{a} | Y^{\text{T}} = y_q) \\ 0 & \text{otherwise} \end{cases},$$

where $\mathbb{E}_{\mathcal{D}}(\tilde{a} | Y^{\text{T}} = y_q)$ is the expectation of CLIP output $\tilde{a}$ corresponding to the description $a'$, for samples in the downstream training set $\mathcal{D}$ whose group truth labels are $y_q$.

For $\omega^{\mathrm{avg}}$, we have:

$$\omega_{p,q}^{\mathrm{avg}} = \hat{p}(V = a_p | Y^{\mathrm{T}} = y_q) = \begin{cases} \frac{1}{m} & a_p \in \mathcal{A}(y_q) \\ 0 & \text{otherwise} \end{cases},$$

where $m = |\mathcal{A}(y_q)|$ is the number of descriptions for each class $y_q$. It can be observed that both $\max(\cdot)$ and $\mathrm{avg}(\cdot)$ are special cases of PRM, which is consistent with Eq. (4) in Section 3. Therefore, $\omega^{\mathrm{max}}$ and $\omega^{\mathrm{avg}}$ exist within the hypothesis space of PRM, meaning:

$$\omega^{\mathrm{max}} \in \Phi^{\mathrm{prm}}, \text{ and } \omega^{\mathrm{avg}} \in \Phi^{\mathrm{prm}}.$$

From Eq. (15), we have:

$$\hat{R}_{\mathcal{D}}^{\mathrm{dvp}}(\Delta, \omega^{\mathrm{max}}) \geq \hat{R}_{\mathcal{D}}^{\mathrm{dvp}}(\Delta, \omega^{\mathrm{prm}*}), \text{ and } \hat{R}_{\mathcal{D}}^{\mathrm{dvp}}(\Delta, \omega^{\mathrm{avg}}) \geq \hat{R}_{\mathcal{D}}^{\mathrm{dvp}}(\Delta, \omega^{\mathrm{prm}*})$$
$$\Rightarrow \hat{R}_{\mathcal{D}}^{\mathrm{dvp}}(\Delta, \omega^{\mathrm{fix}}) \geq \hat{R}_{\mathcal{D}}^{\mathrm{dvp}}(\Delta, \omega^{\mathrm{prm}*}).$$

$\square$

## B.5. Proof of Corollary 4.6

*Proof.* According to Lemma 4.5, $\forall \Delta, \hat{R}_{\mathcal{D}}^{\mathrm{dvp}}(\Delta, \omega^{\mathrm{fix}}) \geq \hat{R}_{\mathcal{D}}^{\mathrm{dvp}}(\Delta, \omega^{\mathrm{prm}*})$, then we have:

$$\hat{R}_{\mathcal{D}}^{\mathrm{dvp}}(\Delta_1^*, \omega^{\mathrm{fix}}) \geq \hat{R}_{\mathcal{D}}^{\mathrm{dvp}}(\Delta_1^*, \omega^{\mathrm{prm}*}) \geq \hat{R}_{\mathcal{D}}^{\mathrm{dvp}}(\Delta_2^*, \omega^{\mathrm{prm}*}),$$

where $\Delta_1^*$ is the optimized $\Delta$ for $\hat{R}_{\mathcal{D}}^{\mathrm{dvp}}(., \omega^{\mathrm{fix}})$, while $\Delta_2^*$ is the jointly optimized $\Delta$ considering $\omega^{\mathrm{prm}*}$. To avoid confusion, we use $\Delta_1^*$ and $\Delta_2^*$ to represent different $\Delta^*$s in Lemma 4.4 and Corollary 4.6.

Lemma 4.4 proves $\hat{R}_{\mathcal{D}}^{vr}(\delta^*, \omega^{\mathrm{fix}}) \geq \hat{R}_{\mathcal{D}}^{\mathrm{dvp}}(\Delta^*, \omega^{\mathrm{fix}})$, i.e., $\hat{R}_{\mathcal{D}}^{vr}(\delta^*, \omega^{\mathrm{fix}}) \geq \hat{R}_{\mathcal{D}}^{\mathrm{dvp}}(\Delta_1^*, \omega^{\mathrm{fix}})$. Applying Lemma 4.4 in the above equation, it can be obtained that:

$$\hat{R}_{\mathcal{D}}^{vr}(\delta^*, \omega^{\mathrm{fix}}) \geq \hat{R}_{\mathcal{D}}^{\mathrm{dvp}}(\Delta_1^*, \omega^{\mathrm{fix}}) \geq \hat{R}_{\mathcal{D}}^{\mathrm{dvp}}(\Delta_2^*, \omega^{\mathrm{prm}*})$$
$$\Rightarrow \hat{R}_{\mathcal{D}}^{vr}(\delta^*, \omega^{\mathrm{fix}}) \geq \hat{R}_{\mathcal{D}}^{\mathrm{dvp}}(\Delta_2^*, \omega^{\mathrm{prm}*}).$$

That is, $\hat{R}_{\mathcal{D}}^{vr}(\delta^*, \omega^{\mathrm{fix}}) \geq \hat{R}_{\mathcal{D}}^{\mathrm{dvp}}(\Delta^*, \omega^{\mathrm{prm}*})$, as $\Delta^*, \omega^{\mathrm{prm}*}$ are jointly optimzed. $\square$

# C. Appendix 3: More Experimental Results

## C.1. Hyper-parameters

### C.1.1. CHOOSING HYPER-PARAMETERS

*Table 7.* Choosing Appropriate Hyper-parameter $k$ with the zero-shot accuracy

| $k$ | AIRCRAFT | CALTECH | CARS | DTD | ESAT | FLOWERS | FOOD | PETS | SUN | UCF | RESISC | AVG. |
|---|---|---|---|---|---|---|---|---|---|---|---|---|
| 1 | 27.00 | 93.67 | 66.67 | 55.67 | 51.26 | 84.53 | 86.36 | 91.25 | 66.08 | 70.74 | 61.65 | 68.63 |
| 3 | 27.36 | 93.63 | 66.52 | 55.44 | 52.12 | 84.41 | 86.43 | 91.39 | 66.28 | 70.79 | 61.67 | **68.73** |
| 5 | 27.78 | 93.31 | 66.46 | 54.55 | 51.32 | 83.52 | 86.54 | 91.25 | 65.61 | 70.58 | 61.43 | 68.39 |
| 7 | 27.69 | 93.31 | 66.47 | 54.43 | 50.17 | 82.54 | 86.55 | 91.22 | 65.15 | 70.26 | 61.33 | 68.10 |

For both DVP-cse and DVP-cls, the optimal value of hyper-parameter $k$ is determined based on the average zero-shot accuracy on the validation sets of downstream tasks under different $k$ values in $\{1, 3, 5, 7\}$. The results are shown in Table 7. When $k = 3$, the average zero-shot accuracy reached its peak. Therefore, we set $k = 3$ in this study.

For hyper-parameter $v$ in DVP-cls, we selected the value from $v \in \{1, 2, 3\}$. Since the sample size and number of classes vary across datasets, we determined a dataset-specific $v$ using the validation set for each dataset. The detailed results are shown in Table 8.

*Table 8.* Choosing Appropriate Hyper-parameter $v$

| $v$ | AIRCRAFT | CALTECH | CARS | DTD | ESAT | FLOWERS | FOOD | PETS | SUN | UCF | RESISC |
|---|---|---|---|---|---|---|---|---|---|---|---|
| 1 | 38.07 | 87.97 | 56.30 | 87.60 | **96.47** | 89.47 | 81.77 | 88.57 | 66.50 | 77.40 | 85.70 |
| 2 | 41.53 | 89.93 | 59.23 | 93.23 | 96.07 | 91.27 | **83.20** | 89.30 | 69.50 | **82.33** | 88.00 |
| 3 | **45.27** | **90.07** | **61.17** | **95.20** | 93.77 | **93.00** | 83.17 | **90.40** | **71.03** | 82.30 | **89.03** |
| OPTIMAL | 3 | 3 | 3 | 3 | 1 | 3 | 2 | 3 | 3 | 2 | 3 |

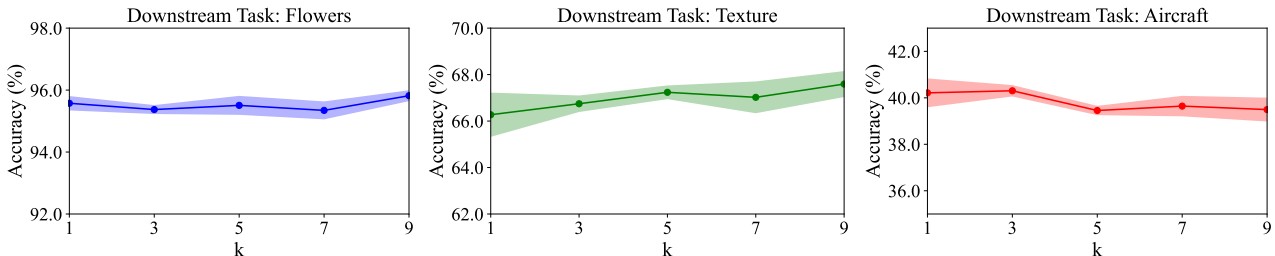

*Figure 8.* Impact of hyper-parameter $k$ on DVP-cse, when $k \in \{1, 3, 5, 7, 9\}$, using ViT-B16-based CLIP as the pretrained model.

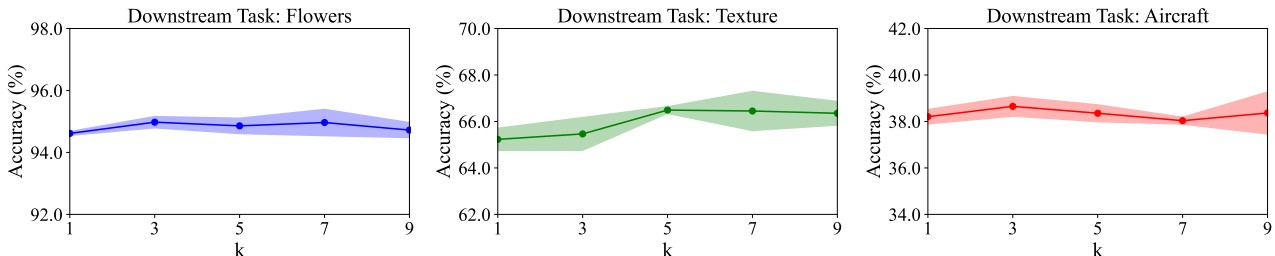

*Figure 9.* Impact of hyper-parameter $k$ on DVP-cls, when $k \in \{1, 3, 5, 7, 9\}$, using ViT-B16-based CLIP as the pretrained model.

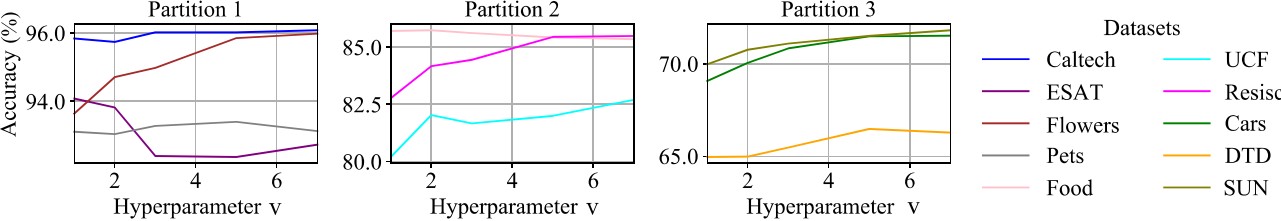

*Figure 10.* Impact of hyper-parameter $v$ on DVP-cls, when $v \in \{1, 2, 3, 5, 7\}$, using ViT-B16-based CLIP as the pretrained model.

### C.1.2. IMPACT OF HYPER-PARAMETERS

**Impact of $k$.** Figures 8 and Figure 9 illustrate the impact of different $k \in \{1, 3, 5, 7, 9\}$ values on DVP-cse and DVP-cls. The $k$ setting is designed to compensate for the limited training data. For datasets with fewer classes and limited training samples, such as the Texture dataset, increasing $k$ within a reasonable range (i.e., k=3, k=5) generally leads to a steady improvement in accuracy. In contrast, for datasets with sufficient samples, such as Flowers and Aircraft, the effect of $k$ is relatively minor.

**Impact of $v$.** Figure 10 shows the impact of different $v$ values on DVP-cls. For most datasets, accuracy increases with $v$ before stabilizing, with a few exceptions. One such exception is the ESAT remote sensing dataset, which contains only 10 relatively simple classes. In this case, a larger $v$, implying more VPs and parameters, may lead to overfitting. Another exception is the Food dataset. As mentioned in Section 5, VR methods generally perform less effectively on this dataset, so increasing $v$ does not yield significant improvements.

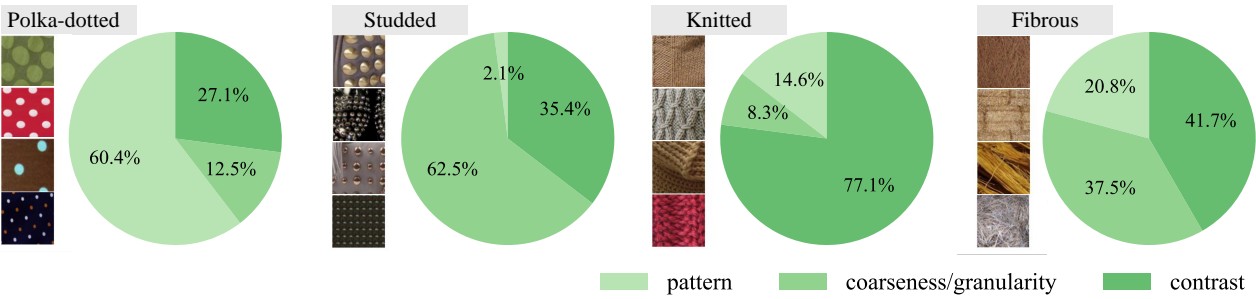

*Figure 11.* Visualization results of DVP-cse, showing weights of causes for classifying certain classes, using ViT-B16-based CLIP.

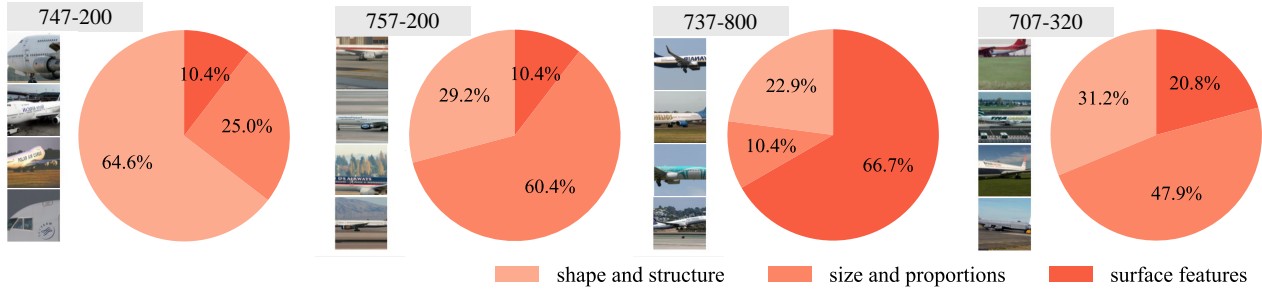

*Figure 12.* Visualization results of DVP-cse, showing weights of causes for classifying certain classes, using ViT-B16-based CLIP.

## C.2. More Visualization Results of DVP-cse

Figure 11 shows the causes' weight distribution for specific classes in the DTD dataset, obtained from the PRM trained with DVP-cse. The results offer a degree of interpretability. For instance, the 'polka-dotted' class, characterized by spotted patterns, has the highest weight for the 'pattern' cause. Similarly, 'studded', usually associated with metallic textures, shows the largest weight for 'coarseness/granularity'. For classes like 'knitted', which are strongly related to weaving, images often exhibit black gaps and solid color areas from the knitting, making contrast the most important classification factor. Meanwhile, classes such as 'fibrous', which have diverse characteristics, exhibit a more balanced distribution of cause weights.

Figure 12 shows the cause weight distribution for specific classes in the Aircraft dataset, obtained from the PRM trained with DVP-cse. The obtained weights also aid in understanding the classification process. For instance, the 747-200, as a double-deck aircraft, has a distinctive shape, making 'shape and structure' the most critical factor. The 757-200, with a longer fuselage and shorter wings compared with other models, is primarily classified based on 'size and proportions'. Meanwhile, the 737-800 typically features two cabin doors and relatively large cabin windows, making 'surface features' a key classification criterion.

## C.3. More Visualization Results of DVP-cls

Figures 13 and Figure 14 illustrate the classes closest to the center of each description cluster in DVP-cls (i.e., sorted by the sum of each sample's distance to the cluster center). As discussed in Section 5, DVP-cls tends to group descriptions of classes with similar attributes–which might belong to the same subclass–into the same cluster.

Figure 13 presents the results on the DTD dataset, where cluster 1 corresponds to overlayed or layered patterns, such as 'interlaced', 'frilly', and 'swirly'. Cluster 2 represents textures characterized by spots, including 'bumpy' and 'flecked'.

Top Classes within Different Clusters (Downstream Task: Texture; Method: DVP-cls)

| Cluster One | Cluster Two | Cluster Three |
|---|---|---|
| interlaced 7.70 | bumpy 5.87 | interlaced 6.19 |
| frilly 7.72 | flecked 6.08 | woven 6.52 |
| swirly 7.81 | blotchy 6.09 | crosshatched 6.98 |
| Sorted by Distance to the Cluster Center | Sorted by Distance to the Cluster Center | Sorted by Distance to the Cluster Center |

*Figure 13.* Visualization results of DVP-cls, showing the top 3 classes closest to the description cluster centers, using ViT-B16-based CLIP.

Top Classes within Different Clusters (Downstream Task: Aircraft; Method: DVP-cls)

| Cluster One | Cluster Two | Cluster Three |
|---|---|---|
| Model B200 8.08 | MD-87 8.10 | 767-200 5.66 |
| DHC-8-300 8.21 | ERJ 135 8.12 | 767-300 5.83 |
| DC-6 8.28 | Fokker 70 8.29 | 767-400 6.17 |
| Sorted by Distance to the Cluster Center | Sorted by Distance to the Cluster Center | Sorted by Distance to the Cluster Center |

*Figure 14.* Visualization results of DVP-cls, showing the top 3 classes closest to the description cluster centers, using ViT-B16-based CLIP.

Cluster 3 relates to interwoven patterns, such as 'interlaced', 'woven', and 'crosshatched'. Figure 14 shows the results on the Aircraft dataset, with consistent conclusions. For example, cluster 3 groups all 767-x models into a single cluster. This clustering effectively reflects the subclass-based organization of different labels.

### C.4. More Results on Different Backbones

*Table 9.* Accuracy comparison of different methods trained on 16-shot downstream classification tasks, using RN50-based CLIP as the pretrained model (Mean %, ours are  highlighted  and the highest is in **bold**).

| METHOD | AIRCRAFT | CALTECH | CARS | DTD | ESAT | FLOWERS | FOOD | PETS | SUN | UCF | RESISC | AVG. |
|---|---|---|---|---|---|---|---|---|---|---|---|---|
| VP | 16.2 | 80.1 | 44.0 | 43.4 | 59.7 | 53.6 | 65.3 | 77.2 | 48.8 | 52.0 | 47.7 | 53.5 |
| AR | 18.6 | 86.5 | 53.9 | 46.4 | 66.6 | 60.9 | 74.2 | 82.5 | 56.8 | 59.7 | 58.4 | 60.4 |
| AttrVR | 20.7 | 89.1 | 53.9 | 54.4 | 72.0 | 74.8 | **75.3** | 88.9 | 59.9 | 63.6 | 58.2 | 64.6 |
| DVP-cse | **23.5** | **89.8** | **55.9** | **57.6** | **74.4** | **81.4** | 74.8 | 88.5 | 60.5 | 65.5 | 60.3 | **66.6** |
| DVP-cls | 22.1 | 89.8 | 54.5 | 55.9 | 72.2 | 80.0 | 75.0 | **88.9** | **61.1** | **65.9** | **60.8** | 66.0 |

Tables 9 to 11 present the detailed results of various VR methods on different datasets when CLIP adopts different image encoder architectures (RN50, RN101, ViT-B32). On average, DVP-cse outperforms the baseline method AttrVR by 1.5% to 2.4%, while DVP-cls, though slightly less effective than DVP-cse, still achieves an average improvement of 1.2% to 1.6% over AttrVR. Notably, the performance gains from DVP are more pronounced when the pretrained image encoder uses simpler architectures, such as RN50 and RN101. The only exception among these datasets is Food. As discussed in Section 5, VR methods are less effective on this dataset, and thus DVP's decoupling-and-reweighting framework does not improve accuracy for this task.

*Table 10.* Accuracy comparison of different methods trained on 16-shot downstream classification tasks, using RN101-based CLIP as the pretrained model (Mean %, ours are highlighted and the highest is in **bold**).

| METHOD | AIRCRAFT | CALTECH | CARS | DTD | ESAT | FLOWERS | FOOD | PETS | SUN | UCF | RESISC | AVG. |
|---|---|---|---|---|---|---|---|---|---|---|---|---|
| VP | 19.3 | 83.0 | 53.7 | 43.4 | 62.8 | 57.2 | 71.2 | 80.2 | 53.5 | 54.2 | 54.0 | 57.5 |
| AR | 19.5 | 89.7 | 62.0 | 46.3 | 70.4 | 60.4 | 78.0 | 84.4 | 58.4 | 60.6 | 60.2 | 62.7 |
| ATTRVR | 23.3 | 92.0 | 62.2 | 55.6 | 70.3 | 76.2 | **79.5** | 89.3 | 62.1 | 64.5 | 64.5 | 67.2 |
| DVP-CSE | **24.7** | **93.3** | **64.1** | **59.2** | **73.3** | **82.9** | 79.3 | **90.4** | 62.4 | 67.6 | 67.8 | **69.6** |
| DVP-CLS | 23.8 | 92.7 | 62.5 | 58.0 | 70.7 | 80.6 | 79.1 | 89.5 | **63.7** | **68.1** | **68.4** | 68.8 |

*Table 11.* Accuracy comparison of different methods trained on 16-shot downstream classification tasks, using ViT-B32-based CLIP as the pretrained model (Mean %, ours are highlighted and the highest is in **bold**).

| METHOD | AIRCRAFT | CALTECH | CARS | DTD | ESAT | FLOWERS | FOOD | PETS | SUN | UCF | RESISC | AVG. |
|---|---|---|---|---|---|---|---|---|---|---|---|---|
| VP | 24.3 | 92.3 | 58.6 | 54.9 | 85.9 | 71.2 | 75.0 | 86.8 | 61.0 | 67.3 | 73.9 | 68.3 |
| AR | 21.8 | 92.7 | 56.9 | 49.9 | 85.6 | 66.7 | 75.7 | 84.7 | 59.9 | 63.5 | 71.6 | 66.3 |
| ATTRVR | 24.5 | 92.0 | 56.6 | 56.8 | 88.6 | 77.8 | **77.2** | 89.8 | 62.8 | 67.9 | 73.9 | 69.8 |
| DVP-CSE | **27.3** | **93.4** | **58.1** | **58.8** | **89.1** | **83.1** | 76.8 | **89.8** | 63.5 | 69.0 | 75.4 | **71.3** |
| DVP-CLS | 26.1 | 92.9 | 56.5 | 57.2 | 88.5 | 82.5 | 77.0 | 89.2 | **64.2** | **70.5** | **76.0** | 71.0 |

## C.5. Discussion about Training Time

*Table 12.* Accuracy comparison of AttrVR with DesAttr and our methods with only one VP trained, using ViT-B16-based CLIP as the pretrained model (Mean %, ours are highlighted and the highest is in **bold**).

| | AIRCRAFT | CALTECH | CARS | DTD | ESAT | FLOWERS | FOOD | PETS | SUN | UCF | RESISC | AVG. |
|---|---|---|---|---|---|---|---|---|---|---|---|---|---|
| ATTRVR (DESATTR) | 35.9 | 95.6 | 68.2 | 64.4 | 93.8 | 92.4 | 85.7 | 93.0 | 67.7 | 78.6 | 81.8 | 77.9 |
| DVP-CSE (NUM=1) | **37.2** | **96.0** | **70.1** | **66.3** | 93.8 | **93.8** | 85.6 | **93.3** | 68.4 | 79.7 | 81.7 | **78.7** |
| DVP-CLS (NUM=1) | 36.4 | 95.8 | 69.1 | 65.3 | **94.1** | 93.6 | **85.7** | 93.1 | **70.0** | **80.2** | **82.8** | **78.7** |

*Table 13.* Training cost of different VR methods, using the ViT-B16-based CLIP as the pretrained model and the Flowers task as an example (ours are highlighted ).

| | VP | AR | ATTRVR | DVP-CSE (NUM=1) | DVP-CLS (NUM=1) |
|---|---|---|---|---|---|
| PARAMETER NUMBER | 69.8K | 39.9K | 39.9K | 39.9K | 39.9K |
| TRAINING TIME FOR EACH EPOCH (S) | 2.97±0.02 | 2.85±0.03 | 2.83±0.03 | 2.87±0.02 | 2.88±0.04 |
| TRAINING TIME IN TOTAL (MIN) | 9.78±0.07 | 9.44±0.05 | 9.54±0.04 | 9.70±0.03 | 9.66 ± 0.07 |

A potential limitation of DVP is the increased training time required due to the larger number of VPs that need to be optimized. However, since each VP is trained independently, the process can be parallelized, minimizing any significant increase in overall training time. Additionally, the time cost of computing the PRM primarily involves matrix operations, which do not require gradient calculations. Compared with the time-consuming forward pass of CLIP, this overhead is negligible.

Even under the constraint that DVP must use only one VP as the baseline method for a fair comparison, it can still achieve improvements over the baseline with almost no additional time overhead. Table 12 compares AttrVR and DVP when only a single VP is used, while Table 13 shows the runtime of various VR methods. For fairness, the number of descriptions is kept consistent, and only one VP is used for each method. The results demonstrate that both DVP-cse and DVP-cls can deliver an average accuracy improvement with negligible extra computational cost.

## C.6. Exploring Impact of Different Causes

Figure 15 demonstrates the impact of using a single cause in DVP-cse on the classification results for Flowers. Some causes, such as flower structure, size, and leaf characteristics, significantly contribute to the success of reprogramming. These causes are well-suited for reprogramming because they are visually distinctive, making them easier to optimize through

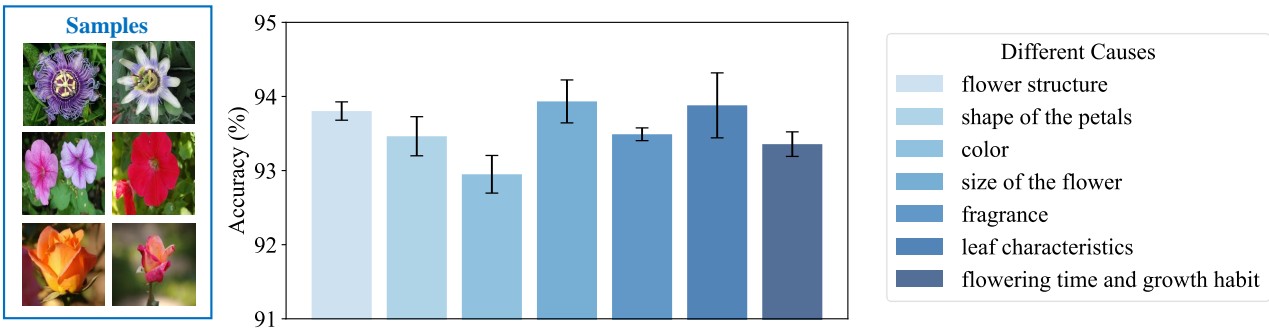

*Figure 15.* Comparison results of different causes for classifying Flowers applying DVP-cse with a single cause, using ViT-B16-based CLIP.

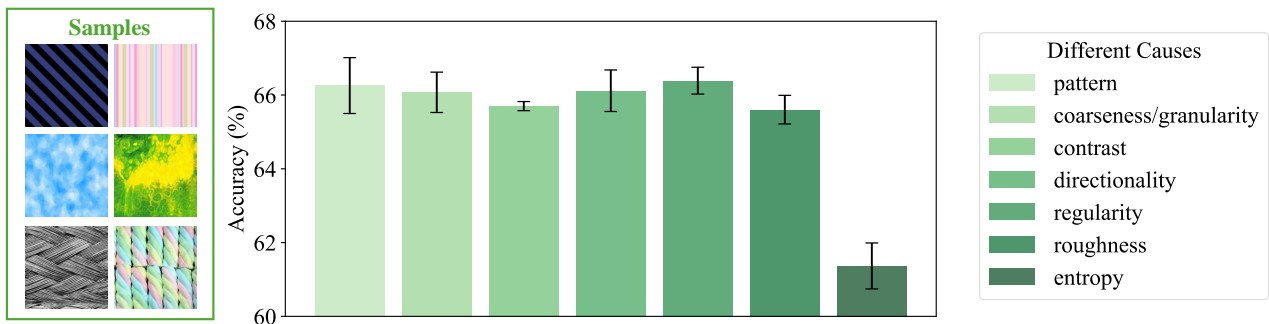

*Figure 16.* Comparison results of different causes for classifying DTD applying DVP-cse with a single cause, using ViT-B16-based CLIP.

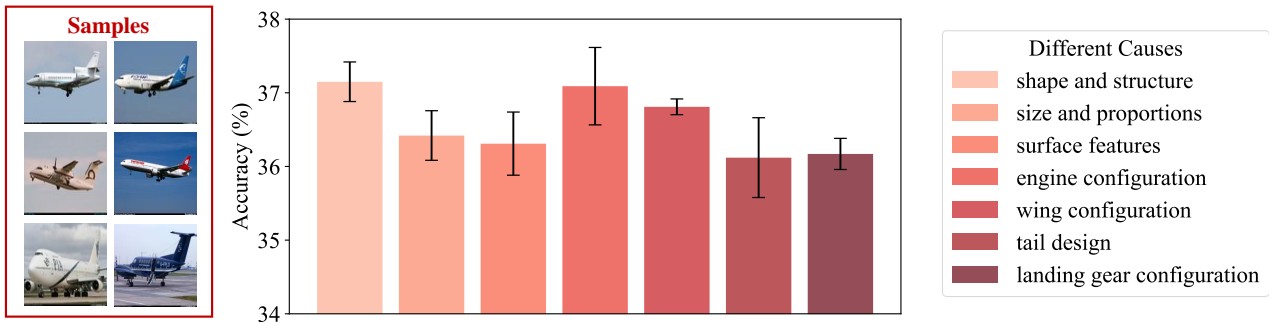

*Figure 17.* Comparison results of different causes for classifying Aircraft applying DVP-cse with a single cause, using ViT-B16-based CLIP.

VP patterns, and they are causally linked to correct classification. Conversely, causes like color, fragrance, and flowering time/growth habit are less effective for reprogramming. Color is less suitable because individual flowers within the same class may exhibit color variations, reducing its contribution to accurate classification. Meanwhile, fragrance and flowering

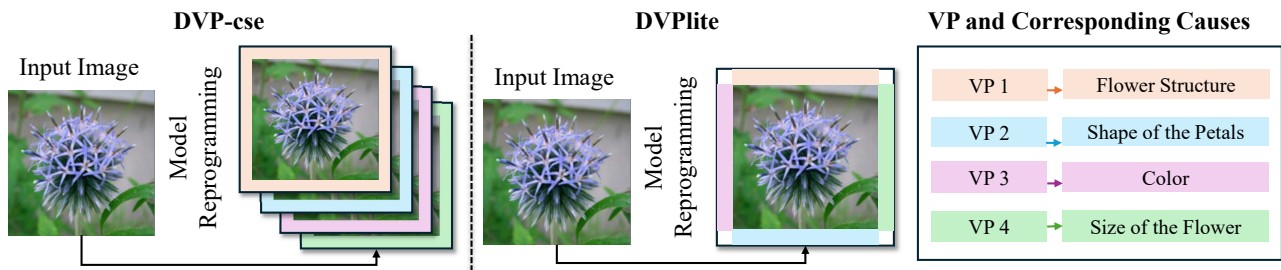

*Figure 18.* Pipeline of the original DVP-cse and DVPlite under the setting of four decoupled causes, using the Flowers dataset as an example. For different causes, DVP-cse requires training multiple separate VPs, multiplying the parameters. In contrast, DVPlite decouples the VPs into up, down, left, and right directions, assigning them to different causes, which maintains parameter count comparable to other VR methods.

time/growth habit are not visual features, making them inherently unsuitable for VR methods.

Figure 16 and Figure 17 respectively illustrate the influence of different causes on classification results for the DTD and Aircraft datasets. In Figure 16, it is evident that most causes, except for entropy, yield similar results, while entropy leads to noticeably poorer VR performance. This is because entropy, as a cause, lacks a direct connection to visual information, making it difficult to optimize classification through VR. In contrast, causes such as pattern, contrast, and regularity are directly tied to visual features, facilitating effective reprogramming. Figure 17 highlights that the most suitable causes for VR in the Aircraft dataset are shape and structure, engine configuration, and wing configuration.

In summary, the most effective attribute descriptions for guiding VR meet two key criteria: (1) they are related to visual information, and (2) they have a causal relationship with accurate classification.

### C.7. DVPlite: A Potential Method for Parameter Reduction

When aiming to enhance the accuracy as DVP does without increasing the parameter overhead, we propose DVPlite as a potential alternative. It enables decoupling-and-reweighting while keeping the parameter number consistent with that of other VR methods.

VR pattern consists of trainable parameters arranged alongside the four edges of the image (i.e., padding). In DVP, each VR pattern is trained independently, resulting in a substantial increase in the total number of parameters. The proposed DVPlite method in this section divides the VR pattern into four independent parts, with a total number of trainable parameters equal to that of AttrVR. Each part is positioned at the top, bottom, left, and right peripheries, as is shown in Figure 18.

*Table 14.* Accuracy comparison of our DVPlite and the best-performing baseline method AttrVR trained on 16-shot downstream classification task, using ViT-B16-based CLIP as the pretrained model (Mean %, ours is highlighted and the highest is in **bold**).

| | AIRCRAFT | CALTECH | CARS | DTD | ESAT | FLOWERS | FOOD | PETS | SUN | UCF | RESISC | AVG. |
|---|---|---|---|---|---|---|---|---|---|---|---|---|
| ATTRVR (BASELINE) | 36.6 | 95.7 | 68.3 | 65.6 | **93.8** | 92.9 | **85.9** | 93.3 | 69.6 | 79 | 82.6 | 78.5 |
| DVPLITE | **39.3** | **95.9** | **71.4** | **66.5** | **93.8** | **95.2** | 85.8 | **93.4** | **71.6** | **81** | **83.6** | **79.8** |

*Table 15.* A comparison of parameter numbers for different methods and their average accuracy across 11 datasets, using ViT-B16-based CLIP as the pretrained model (Mean %, ours are highlighted .

| | VP | AR | ATTRVR | DVP-CSE | DVP-CLS | DVPLITE |
|---|---|---|---|---|---|---|
| PARAMETER NUMBERS | 0.07M | 0.04M | 0.04M | 0.12M | 0.04-0.12M | 0.04M |
| AVERAGE ACCURACY OVER 11 TASKS | 74.4 | 76.5 | 78.5 | 80.1 | 79.7 | 79.8 |

Accuracy comparisons of our DVPlite and the best-performing baseline method AttrVR are shown in Table 14, and the parameter numbers are shown in Table 15. In the DVPlite experiments presented here, the assignment of which specific VP was applied to which spatial location was determined randomly (and kept fixed for that run). DVPlite is then implemented

under our proposed two-stage DVP-PRM framework, with the four parts trained independently.

Therefore, the core DVP-PRM principles yield benefits without increasing the parameter count. DVPlite outperformed the AttrVR baseline with an identical parameter budget, demonstrating the effectiveness of the decoupling-and-reweighting strategy itself. However, investigating structured or learnable assignments of VPs to spatial locations instead of random assignments might be an interesting open question that deserves formal and systematic exploration in future studies.

### C.8. Applying DVP-PRM to Different VR methods

*Table 16.* Results of applying our DVP-PRM framework to various VR methods, including VP and AR (for AR, results equal to that of DVP-cse and DVP-cls), trained on a 16-shot downstream classification task, using ViT-B16-based CLIP as the pretrained model.

|  | AIRCRAFT | CALTECH | CARS | DTD | ESAT | FLOWERS | FOOD | PETS | SUN | UCF | RESISC | AVG. |
|---|---|---|---|---|---|---|---|---|---|---|---|---|
| VP + OURS-CSE (%) | + 6.2 | + 1.2 | + 4.3 | + 4.0 | + 0.6 | + 9.2 | + 0.2 | + 1.1 | + 2.4 | + 2.6 | + 2.4 | + 3.1 |
| VP + OURS-CLS (%) | + 6.3 | + 0.8 | + 1.2 | + 2.0 | + 0.0 | + 8.4 | + 0.2 | + 1.1 | + 2.5 | + 3.5 | + 2.7 | + 2.6 |
| AR + OURS-CSE (%) | + 8.6 | + 0.7 | + 4.5 | + 4.7 | + 0.5 | + 9.5 | + 0.4 | + 0.4 | + 3.2 | + 3.6 | + 3.0 | + 3.6 |
| AR + OURS-CLS (%) | + 7.0 | + 0.5 | + 2.8 | + 3.5 | + 0.7 | + 9.1 | + 0.5 | + 0.6 | + 3.2 | + 3.9 | + 2.8 | + 3.1 |

Given the existence of various VR methods, including watermarking-based (i.e. VP(Bahng et al., 2022)) and padding-based (i.e. AR (Tsai et al., 2020)), we apply the proposed DVP-PRM framework to both types to evaluate its performance. Notably, applying the framework to the padding-based VR setting corresponds to the DVP-cse and DVP-cls results presented in this paper. Performance improvements achieved by incorporating our module into different VR methods are summarized in Table 16. These results suggest that our method is effective for both watermarking-based and padding-based VR, consistently providing performance gains across these different approaches.

### C.9. Training Results Under Sparse Data Conditions

*Table 17.* Accuracy comparison of different VR methods with fewer or more training samples (i.e., 1-, 4-, 8-, and 32-shot tasks), using ViT-B16-based CLIP as the pretrained model and the Aircraft task as an example (Mean %±Std %, ours are highlighted and the highest is in **bold**).

|  | VP | AR | ATTRVR | DVP-CSE | DVP-CLS |
|---|---|---|---|---|---|
| 1-SHOT | $24.1_{\pm 0.4}$ | $25.1_{\pm 0.4}$ | $\mathbf{31.5}_{\pm 0.2}$ | $28.1_{\pm 0.8}$ | $29.4_{\pm 0.5}$ |
| 4-SHOT | $27.5_{\pm 0.2}$ | $27.7_{\pm 0.4}$ | $\mathbf{33.4}_{\pm 0.6}$ | $32.3_{\pm 0.1}$ | $32.5_{\pm 0.3}$ |
| 8-SHOT | $29.5_{\pm 0.3}$ | $30.1_{\pm 0.1}$ | $35.1_{\pm 0.3}$ | $\mathbf{36.4}_{\pm 0.1}$ | $35.9_{\pm 0.5}$ |
| 16-SHOT | $32.1_{\pm 0.6}$ | $31.7_{\pm 0.3}$ | $36.6_{\pm 0.3}$ | $\mathbf{40.3}_{\pm 0.2}$ | $38.7_{\pm 0.4}$ |
| 32-SHOT | $34.5_{\pm 0.4}$ | $34.2_{\pm 0.7}$ | $38.4_{\pm 0.3}$ | $\mathbf{43.1}_{\pm 0.8}$ | $41.2_{\pm 0.5}$ |

To evaluate the performance of DVP under varying amounts of training data, we conducted experiments on the Aircraft dataset with 1-, 4-, 8-, 16-, and 32-shot settings. The results are presented in Table 17[2].

It can be observed that under extremely limited training conditions, such as the 1-shot and 4-shot scenarios, DVP performs worse than the baseline methods. This is attributed to the fact that the MLE in the PRM module becomes more reliable with larger amounts of data. The suboptimal performance under very low-data conditions highlights one of the limitations of DVP.

However, the performance improvement of DVP becomes increasingly evident when the number of training shots exceeds 8 for the Aircraft dataset, and is particularly pronounced under the 32-shot setting. This suggests that as the amount of training data increases, DVP can better leverage its modeling capacity, resulting in more substantial gains.

## D. Appendix 4: Problem Setup & Notation

### D.1. Problem Setup Comparison

We clarify the differences between the problem this work focuses on and another active research theme to avoid confusion.

---

[2]Note that we conducted additional experiments and correct the 8-shot and 32-shot results reported in the rebuttal, but these corrections do not affect the conclusions presented during rebuttal.

**Visual Reprogramming (This Work).** The focus of this paper is to address the challenge of adapting CLIP, pretrained on aligned image-text pairs, to downstream classification tasks where input dimensions mismatch the original model ($d_T \neq d_S$). Following established visual reprogramming protocols, we learn a set of visual prompts (VPs) $\Delta = \delta_1, \ldots, \delta_v$ that perturb input images to align them with CLIP's pretrained feature space. Formally, given a downstream dataset $\mathcal{D} = \{(x_j^T, y_j^T)\}_{j=1}^N$ with images $x_j^T \in \mathcal{X}^T$ and labels $y_j^T \in \mathcal{Y}^T$, we optimize $\Delta$ to minimize the cross-entropy loss:

$$\Delta^* = \arg\min_\Delta \left( -\frac{1}{N} \sum_{j=1}^N \log p(y_j^T | f_{\text{in}}(x_j^T; \Delta), \mathcal{A}) \right).$$

where $f_{\text{in}}(x^T; \Delta) = \texttt{pad}(x^T) + \Delta$ pads to match CLIP's expected dimensions and applies *trainable visual prompts* $\Delta$.

**Text Prompt Tuning.** In contrast, text prompt tuning methods like CoOp (Zhou et al., 2022a) adapt pretrained CLIP by modifying its *text* input space. Instead of perturbing images, these studies aim to optimize a set of continuous context vectors $\mathbf{C} = [\mathbf{c}_1, \ldots, \mathbf{c}_M]$ prepended to class names, forming prompts such as "$\mathbf{c}_1 \cdots \mathbf{c}_M$ [Class Info]." For class $y$, its embedding becomes

$$f_{\text{txt}}(\mathbf{C}, y) = f_{\text{txt}}\left(\texttt{concat}(\mathbf{c}_1, \ldots, \mathbf{c}_M, f_{\text{tokenize}}(y))\right).$$

where $f_{\text{tokenize}}(y)$ tokenizes the class name $y$. The learning objective maximizes the alignment between image embeddings $f_{\text{img}}(x^T)$ and adapted text embeddings $f_{\text{txt}}(\mathbf{C}, y^T)$.

$$\mathbf{C}^* = \arg\min_\mathbf{C} \left( -\frac{1}{N} \sum_{j=1}^N \log p(y_j^T | x_j^T; \mathbf{C}) \right).$$

## D.2. Notations

We then provide a list of key mathematical notations used throughout the paper to ensure clarity and consistency.

### D.2.1. GENERAL CONCEPTS REGARDING CLIP MODEL

*Table 18.* General Concepts and Notation

| Symbol | Description |
|---|---|
| CLIP | Contrastive Language-Image Pre-training model (Radford et al., 2021). |
| $f_{\text{img}}$ | CLIP's image encoder. |
| $f_{\text{txt}}$ | CLIP's text encoder. |
| $\mathcal{X}^S$ | Input image space for the pre-trained CLIP model (i.e., source domain images). $\mathcal{X}^S \subseteq \mathbb{R}^{d_S}$. |
| $\mathcal{V}$ | Text space containing all textual descriptions. |
| $\mathcal{Z}$ | Shared embedding space for images and text. |
| $V$ | A textual phrase or description from $\mathcal{V}$. |
| $\mathcal{X}^T$ | Input image space for a downstream task (target domain images). $\mathcal{X}^T \subseteq \mathbb{R}^{d_T}$. |
| $\mathcal{Y}^T$ | Label space for the downstream task. |
| $x^T$ | An image from the target domain $\mathcal{X}^T$. |
| $y^T$ | A label from the target label space $\mathcal{Y}^T$. |
| $d_S$ | Dimensionality of source domain images. |
| $d_T$ | Dimensionality of target domain images. |
| $f_{\text{clip}}(x^S, V)$ | CLIP similarity score between image $x^S$ and text $V$. Defined in Eq. (1). |
| $\tau$ | Temperature parameter in CLIP's similarity computation. |
| $\mathcal{A}$ | The complete set of textual descriptions used for all classes in a downstream task. |
| $\mathcal{A}(y^T)$ | Subset of $\mathcal{A}$ containing $m$ descriptions specifically for class $y^T$. |
| $m$ | Number of textual descriptions per class. |
| $[f_{\text{logits}}(x^T; \mathcal{A})]_{y^T}$ | Logit for class $y^T$ given image $x^T$, computed using standard CLIP when $d_T = d_S$. |
| $\text{agg}(\cdot)$ | Generic aggregation function (e.g., $\texttt{max}(\cdot)$, $\texttt{avg}(\cdot)$) over description similarities. |
| $|\cdot|$ | The cardinality of a set, i.e., the number of elements in a set. |

## D.2.2. STANDARD VISUAL REPROGRAMMING (VR)

*Table 19.* Standard Visual Reprogramming Notation

| Symbol | Description |
|---|---|
| $f_{\text{in}}(x^{\text{T}}\|\delta)$ | Input transformation function in VR, $f_{\text{in}}(x^{\text{T}}\|\delta) = \text{pad}(x^{\text{T}}) + \delta$. |
| $\text{pad}(\cdot)$ | Zero-padding function to make $x^{\text{T}}$ dimensionally compatible with $f_{\text{img}}$. |
| $\delta$ | The single, trainable visual prompt (a vector or tensor in $\mathbb{R}^{d_{\text{S}}}$) in standard VR. |
| $[f^{\text{vr}}_{\text{logits}}(x^{\text{T}}; \delta, \mathcal{A})]_{y^{\text{T}}}$ | Logit for class $y^{\text{T}}$ given image $x^{\text{T}}$ and prompt $\delta$ in standard VR. Defined in Eq. (1). |
| $p_{\text{vr}}(y^{\text{T}}\|x^{\text{T}}; \delta, \mathcal{A})$ | Normalized probability for class $y^{\text{T}}$ in standard VR, obtained via softmax over $f^{\text{vr}}_{\text{logits}}$. |
| $M_a$ | Row vector $(1 \times \|\mathcal{A}\|)$ of CLIP similarities between $f_{\text{in}}(x^{\text{T}}\|\delta)$ and all descriptions in $\mathcal{A}$. |
| $M_y$ | Row vector $(1 \times \|\mathcal{Y}^{\text{T}}\|)$ of class logits. |
| $\omega$ | General reweighting matrix $(\|\mathcal{A}\| \times \|\mathcal{Y}^{\text{T}}\|)$ relating $\mathbf{a}^{\top}_{x,\delta}$ to $\mathbf{y}^{\top}_{x,\delta,\omega}$ via $\mathbf{a}^{\top}_{x,\delta}\,\omega = \mathbf{y}^{\top}_{x,\delta,\omega}$. |
| $\omega_{p,q}$ | Element of $\omega$, representing the contribution of the $p$-th description to the $q$-th class logit. |
| $\omega^{\text{fix}}$ | Fixed reweighting matrix used in standard VR, determined by the choice of $\text{agg}(\cdot)$ (e.g., $\text{max}(\cdot)$ or $\text{avg}(\cdot)$). |
| $\mathcal{D}$ | Downstream training set, consisting of $N$ image-label pairs: $\{(x^{\text{T}}_j, y^{\text{T}}_j)\}^{N}_{j=1}$. |
| $N$ | Total number of samples in the training set $\mathcal{D}$. |

## D.2.3. DECOUPLED VISUAL PROMPTS (DVP) FRAMEWORK

*Table 20.* Decoupled Visual Prompts (DVP) Framework Notation

| Symbol | Description |
|---|---|
| $\Delta$ | Set of $v$ visual prompts, i.e., $\Delta = \{\delta_1, \delta_2, \ldots, \delta_v\}$. |
| $\delta_i$ | The $i$-th decoupled visual prompt, specialized for description partition $\mathcal{A}_i$. |
| $\mathcal{A}$ | The full description set. |
| $\mathcal{E}$ | The full set of text embeddings for descriptions in $\mathcal{A}$. |
| $\mathcal{A}_i$ | The $i$-th partition of the full description set $\mathcal{A}$. |
| $\mathcal{E}_i$ | The set of text embeddings corresponding to descriptions in partition $\mathcal{A}_i$. |
| $v$ | Number of decoupled visual prompts and description partitions. |
| DVP-cse | DVP variant where partitions $\mathcal{A}_i$ are formed based on explicit semantic causes from an LLM. |
| DVP-cls | DVP variant where partitions $\mathcal{A}_i$ are formed by unsupervised clustering of description embeddings. |
| PRM ($\omega^{\text{prm}}$) | Probabilistic Reweighting Matrix: The learned $(\|\mathcal{A}\| \times \|\mathcal{Y}^{\text{T}}\|)$ matrix in the DVP framework. |
| $\omega^{\text{prm}}_{p,q}$ | Entry of PRM, estimated as $\hat{p}(V = a_p\|Y^{\text{T}} = y_q, V \in \mathcal{A}(y_q))$. |
| $\mathcal{N}_{\mathcal{D}}(a, y)$ | Co-occurrence count of description $a$ and class $y$ in training set $\mathcal{D}$. Used for PRM estimation. |
| $\mathcal{K}(x^{\text{T}}_j, k)$ | Set of top-$k$ descriptions in $\mathcal{A}$ most similar to $f_{\text{in}}(x^{\text{T}}_j\|\Delta)$ for image $x^{\text{T}}_j$. |
| $k$ | Number of top similar descriptions considered for smoothed PRM counting. |
| $p_{\text{vr}}(y^{\text{T}}\|x^{\text{T}}; \delta_i, \omega^{\text{prm}}, \mathcal{A}_i)$ | Probability derived from $[f^{\delta_i}_{\text{logits}}(x^{\text{T}})]$. |
| $[f^{\text{dvp}}_{\text{logits}}(x^{\text{T}}; \Delta, \omega^{\text{prm}})]_{y^{\text{T}}_q}$ | Integrated logit for class $y^{\text{T}}_q$ in DVP, combining all prompts $\Delta$ and PRM $\omega^{\text{prm}}$. |
| $p_{\text{dvp}}(y^{\text{T}}\|x^{\text{T}}; \Delta, \omega^{\text{prm}}, \mathcal{A})$ | Final probability for class $y^{\text{T}}$ in DVP, from integrated logits. |

