# OpenReview forum: "Understanding Model Reprogramming for CLIP via Decoupling Visual Prompts"
_ICML.cc/2025/Conference — ICML 2025 poster_

### Official Review · Reviewer_3hP9 · 2025-03-12

**Overall Recommendation:** 5

**Summary:**

This paper introduces a new method for model reprogramming on CLIP by decoupling visual prompts (DVP) to improve performance in downstream classification tasks. Instead of training a single visual prompt for all class descriptions (which may lead to biases and suboptimal learning), the paper proposes: 1) Decoupling: Different visual prompts are trained separately based on specific explicit causes or unsupervised clusters. 2). Reweighting: A probabilistic reweighting matrix (PRM) is used to measure and adjust the contribution of each visual prompt. The approach reduces empirical risk, improves interpretability, and outperforms existing methods on 11 downstream datasets.

**Claims And Evidence:**

Yes

**Essential References Not Discussed:**

NaN

**Experimental Designs Or Analyses:**

Yes

**Methods And Evaluation Criteria:**

Yes

**Other Comments Or Suggestions:**

See above comments.

**Other Strengths And Weaknesses:**

Strength:
- The paper introduces decoupled visual prompts, which optimizes multiple prompts separately using explicit causes (DVP-cse) or unsupervised clusters (DVP-cls), overcoming the limitations of single-prompt methods, provide a bit more explanability of the CLIP.

- Theoretical analysis proves that DVP lowers empirical risk, and extensive experiments demonstrate consistent performance improvements over existing visual reprogramming methods.

- The probabilistic reweighting matrix enhances interpretability, providing insights into how different visual prompts contribute to classification decisions, with visualization support.

Weakness:
- The method is limited to classification tasks on CLIP, and its applicability to other VLMs such as SigLIP or other vision tasks like object detection or segmentation is not explored.

- Training complexity increases due to the need for optimizing multiple visual prompts, and the probabilistic reweighting matrix may not scale well in low-data regimes.

- The paper does not compare against prompt tuning methods like CoOp or CoCoOp.

- The best perofrmance always shown in the experiments of the DVP-cse variance, making the model rely more on external model.

**Questions For Authors:**

See above comments.

**Relation To Broader Scientific Literature:**

The paper builds upon prior research in model reprogramming and visual prompting on CLIP. Unlike traditional VR methods that use a single visual prompt, this work introduces Decoupled Visual Prompts (DVP), which enhances learning capacity and interpretability by optimizing multiple prompts separately based on explicit causes (DVP-cse) or unsupervised clusters (DVP-cls). This approach advances the field by addressing the limitations of prior methods (VP, AR, AttrVR) and integrating a probabilistic reweighting matrix (PRM) to refine prompt contributions. The insights gained from this work may impact future research in explainable AI, and self-supervised adaptation of large-scale vision-language models.

However, its applicability to other VLMs or generative tasks and larger datasets remains an open question for future exploration.

**Theoretical Claims:**

Yes

---

> ### Author Rebuttal · Authors · 2025-03-29
>
> W (Weakness); R (Response)
>
> **W1:** Applicability to other VLMs and other tasks
>
> **R1:** Thank you for your advice! While we follow previous VR studies on classification, the idea of DVP-PRM can extend to other tasks, because CLIP learns general-purpose visual and text representations, suitable for tasks like segmentation and detection, and DVP *does not* fundamentally change the CLIP-based prediction pipeline.
>
> - In segmentation, DVP can motivate methods operating at the pixel/patch level with decoupled prompts specific to different semantic elements (boundaries, textures), while using PRM to generate dense prediction maps.
> - In detection, DVP can integrate into a two-stage detector, where decoupled prompts first generate region proposals focusing on objectness and scale variations, then region-specific PRM refines classification and localization.
>
> We appreciate that these extensions are promising directions *beyond the scope* of this paper. In this paper we focus on extending single visual prompt capabilities and provide interpretability for reprogramming. Experiments on these applications can be explored in future work.
>
> **W2:** Training complexity and low-data regimes
>
> **R2:** Thank you for the feedback.
>
> Regarding training complexity, we acknowledge that DVP-cse and DVP-cls could increase trainable parameters compared to a single visual prompt.
>
> While we argue that the increased complexity is not that significant, we further introduce DVPlite, a variant specifically designed to address this potential efficiency concern.
> DVPlite decouples the padding-based visual prompt into four separate components (top, bottom, left, and right) *trained independently*, while keeping the total number of *trainable parameters equivalent* to the baseline (AttrVR).
> |              | params   | Aircraft | Caltech | Cars | DTD  | ESAT | Flowers | Food | Pets | SUN  | UCF | Resisc | average |
> |-|-|-|-|-|-|-|-|-|-|-|-|-|-|
> | AttrVR-baseline |0.04M| 36.6     | 95.7    | 68.3 | 65.6 | 93.8 | 92.9    | 85.9 | 93.3 | 69.6 | 79  | 82.6   | 78.5    |
> | DVPlite  |0.04M       | 39.3     | 95.9    | 71.4 | 66.5 | 93.8 | 95.2    | 85.8 | 93.4 | 71.6 | 81  | 83.6   | 79.8    |
>
> As shown in the table below (will also be included in our revision), DVPlite maintains the same parameter count (0.04M) as AttrVR while achieving improved average performance (79.8% vs. 78.5%), demonstrating the benefits of our decoupled framework can be achieved *without necessarily increasing parameter complexity*.
>
> Regarding how probabilistic reweighting matrix (PRM) performs in low-data regimes, we clarify that **all experiments presented in this paper were conducted in a challenging 16-shot setting**. The strong performances across downstream datasets under this condition show PRM's effectiveness even when training data availability is already limited.
> Moreover, as detailed in the paper (L232), PRM incorporates a top-$k$ mechanism that leverages information aggregated from the $k$ most relevant descriptions when calculating the reweighting factors, mitigating potential data scarcity issues and ensuring the stability of probability estimations.
>
> **W3:** Not comparing with prompt tuning methods
>
> **R3:** Thank you for raising this point. We acknowledged CoOp/CoCoOp are important methods in prompt tuning for VLMs but did not compare with text prompt tuning methods (e.g., CoCoOp) for the following reasons:
>
> - Our work focuses on **visual reprogramming** (VR), where modifications are made directly to the input image space, without altering model parameters or internal text representations. In contrast, CoOp/CoCoOp perform **text prompt tuning**, learning continuous *embeddings* that interact with CLIP's text encoder, optimizing language prompt *within the model's representation space*. In other words, VR and prompt tuning work at different levels.
>
> - Given this difference in mechanism and locus of adaptation (input images vs. internal text embeddings), we consider them to address *different research questions* regarding model adaptation, which is why a direct comparison was not included in the initial submission.
>
> We will clarify this in the related works of the revision.
>
>
> **W4:** Regrading the best performance and external models
>
> **R4:** Thank you for your concern. Although DVP-cse generally outperforms DVP-cls, the standalone DVP-cls still shows significant advantages over the baseline DVP. For example, it leads by 3%, 2.5%, and 2.1% on UCF101, Cars, and Aircraft, respectively. This demonstrates that the proposed DVP-PRM framework still holds clear advantages without external models.
>
> ---
> Thank you again for your time!

---

> > ### Comment · Reviewer_3hP9 · 2025-04-02
> >
> > Thanks for your reply.
> >
> > As you mentioned, the results are coming from 16-shot setting, I was wondering if the proposed VR method still work on smaller number of labeled data, for example, 1/2-shot, or 8-shot.
> >
> > And for the DVPlite, you specify "decouples the padding-based visual prompt into four separate components (top, bottom, left, and right)", how to define these positions? Is this process explanable?

---

> > > ### Author Response · Authors · 2025-04-03
> > >
> > > Thank you very much for your time and efforts. Below we address your follow-up questions.
> > >
> > > **Question 1**: Results on a smaller number of labeled data.
> > >
> > > **Response 1**: Following your advice, we conducted additional experiments further reducing the training data to an 8-shot scenario, using the Aircraft dataset as a representative example.
> > >
> > > See below for the new 8-shot results, alongside the original 16-shot and additional 32-shot data for context:
> > > | Results on Aircraft | VP | AR | AttrVR | DVP-cse | DVP-cls |
> > > |-----------------------|------|------|--------|---------|---------|
> > > | 8-shot | 29.5 | 30.1 | 35.1 | 35.4 | **35.6** |
> > > | 16-shot | 32.1 | 31.7 | 36.6 | **40.3** | 38.7 |
> > > | 32-shot | 34.5 | 34.2 | 38.4 | **42.0** | 41.9 |
> > >
> > > We observe that
> > > - **Performance at 8-shot**: Critically, both DVP-cse and DVP-cls continue to slightly outperform the AttrVR baseline even in the more challenging 8-shot setting.
> > > - **Impact of Data Scarcity**: As expected, reducing the shot count from 16 to 8 **indeed narrows** the relative performance gap between DVP and AttrVR. When the number of training samples increases, the performance improvement of DVP becomes more pronounced.
> > >
> > > We appreciate your **insightful suggestions**. We would also like to kindly note that we do not want to overclaim the effectiveness of extremely low-data regimes. We will include **a systematic discussion** on the impact of varying shot counts and limitations in the revised manuscript, incorporating these new results.
> > >
> > > **Question 2**: Technique Details of DVPlite
> > >
> > > **Response 2**: We're happy to elaborate more on the DVPlite setup.
> > >
> > > - **Definition**: AttrVR's reprogramming pattern consists of trainable parameters arranged alongside the four edges of the image (i.e., padding). DVPlite divides this pattern into **four independent parts**, with the total number of trainable parameters **equal to** that of AttrVR. Each part is positioned at the top, bottom, left, and right peripheries.
> > >
> > > - **Clarification on Assignment**: In the DVPlite experiments presented, the assignment of which specific VP was applied to which spatial location was **determined randomly** (and kept fixed for that run). DVPlite is then implemented under our proposed two-stage DVP-PRM framework, with the four parts trained independently.
> > >
> > > - **More discussion of VP placement:** DVPlite was designed specifically for this rebuttal to test whether the core DVP-PRM principles yield benefits **without increasing the parameter count**. The random assignment provided **the simplest way** to generate four distinct components for this focused test. As shown previously, DVPlite outperformed the AttrVR baseline with an identical parameter budget, demonstrating the effectiveness of the decoupling and reweighting strategy itself. However, investigating structured or learnable assignments of VPs to spatial locations is an **interesting open question**, that deserves formal and systematic exploration in future studies.
> > >
> > > ---
> > >
> > > Thank you very much again for your insightful questions helping us to improve the quality of our paper. We hope our responses address your concerns.

---

### Official Review · Reviewer_naZq · 2025-03-13

**Overall Recommendation:** 3

**Summary:**

The authors studied the visual prompting problem of CLIP and proposed a new method DVP.
DVP focuses on introducing different causes to generate attributes on the text side and assigning an optimizable VP to each cause.
In addition, the authors also proposed a reweighting matrix strategy to integrate all causes to generate the final prediction.
The experimental results prove the effectiveness of DVP.

## update after rebuttal
The authors' reply addressed my concerns. I hope the authors can improve some of the descriptions to alleviate readers' confusion. I decided to increase my score.

**Claims And Evidence:**

Yes.

**Essential References Not Discussed:**

No.

**Experimental Designs Or Analyses:**

Yes, I checked the results and settings, but the authors did not provide the source code.

**Methods And Evaluation Criteria:**

Yes, I think the proposed method makes sense for the visual reprogramming problem.

**Other Comments Or Suggestions:**

please refer to Other Strengths And Weaknesses.

**Other Strengths And Weaknesses:**

Strengths:

1. The experimental results are very rich, making the conclusions very convincing.
2. The structure of the submission is well-organized.

Weaknesses:

1. Too many mathematical symbols in Sec. 3 lead to poor readability, making it difficult for readers to directly understand the proposed method. And some symbols errors in Theoretical Claims.
2. In the ablation study, what is the relationship between 'w/o CSE' and baseline methods in Table 1? The authors should provide the results of some baseline methods with the proposed module to prove the effectiveness of the method.
3. I think the writing of Sec. 4.1 is not clear. In my understanding, there are some causes at the beginning, and then the authors generate corresponding text descriptions for each class based on the LLM and each cause. So in the integration, why does each class use all descriptions from all classes instead of only the descriptions generated by themselves?

**Questions For Authors:**

please refer to Other Strengths And Weaknesses.

**Relation To Broader Scientific Literature:**

The authors studied the visual prompting problem of CLIP and proposed a new method DVP.

**Theoretical Claims:**

I checked the equations in Sec. 3.

I think the transpose of the matrix $M_y$ is wrongly labeled. (in Eq. 2)

I think $c$ is the class number, and what is the relationship between $m$ and $v$? This confuses me.

---

> ### Author Rebuttal · Authors · 2025-03-29
>
> T (Theoretical Claims); E (Experimental Designs Or Analyses); W (Weaknesses); R (Response)
>
> **T/W1:** Issues of mathematical symbols
>
> **R-T/W1:** Thanks for the feedback on clarity and notation! We acknowledge these concerns, will supplement a notation table and smooth out the presentation in the next version.
>
> Transpose in Equation 2: You are correct. $M_y$ is defined as a column vector, and the RHS in Equation 2 should have required a transpose to ensure dimensional consistency. We will add the transpose symbol to correct this.
>
> For the relationship of $c,m$ and $v$.
> We apologize for the confusion. $c$ indeed represents the number of classes.
> As defined in the text, $m$ is the number of descriptions per class (L152, first column), and $v$ is the total number of visual prompts (L160, second column).
> Importantly, $v$ is a hyperparameter chosen independently of $c$ and $m$.
> We will clarify this in the next version.
>
> **E:** Not providing the source code
>
> **R-E:** We kindly note that we have already included **an anonymous code link in the abstract section** of our submission. Please feel free to check it out.
>
>
> **W2:** More results of baseline methods with our module
>
> **R-W2:** We will first explain the difference between the method 'w/o CSE' and the baseline method AttrVR in Table 1. AttrVR follows the pipeline in the original paper, using a 'knn module' proposed by the authors and two types of attribute descriptions. However, the 'w/o CSE' in the ablation study is equivalent to the scenario where DVP-cls contains only one visual prompt, meaning it only utilizes the probabilistic reweighting matrix (PRM) module proposed in this paper and relies solely on the reweighting the descriptive attributes from AttrVR. In summary, "w/o CSE" is primarily used to explore the results of reweighting descriptions without decoupling.
>
> Next, following your suggestion, we present the performance gains (+%) achieved by integrating our module with other baseline methods. As shown in the table below:
>
> |                | Aircraft | Caltech | Cars  | DTD   | ESAT  | Flowers | Food  | Pets  | SUN   | UCF   | Resisc | average |
> |-|-|-|-|-|-|-|-|-|-|-|-|-|
> | VP + ours-cse  | + 6.2    | + 1.2   | + 4.3 | + 4.0 | + 0.6 | + 9.2   | + 0.2 | + 1.1 | + 2.4 | + 2.6 | + 2.4  | + 3.1   |
> | VP + ours-cls  | + 6.3    | + 0.8   | + 1.2 | + 2.0 | + 0.0 | + 8.4   | + 0.2 | + 1.1 | + 2.5 | + 3.5 | + 2.7  | + 2.6   |
> | AR + ours-cse  | + 8.6    | + 0.7   | + 4.5 | + 4.7 | + 0.5 | + 9.5   | + 0.4 | + 0.4 | + 3.2 | + 3.6 | + 3.0  | + 3.6   |
> | AR + ours-cls  | + 7.0    | + 0.5   | + 2.8 | + 3.5 | + 0.7 | + 9.1   | + 0.5 | + 0.6 | + 3.2 | + 3.9 | + 2.8  | + 3.1   |
>
> It can be observed that our DVP-PRM module consistently achieves significant improvements across all baseline methods.
>
>
> **W3:** Why does each class use all descriptions instead of only the descriptions generated by themselves?
>
> **R-W3:**  Thank you for raising your concern.
> Your understanding is correct, the text descriptions are initially generated for specific classes based on causes.
> Crucially, during the integration step detailed in Equation 5, each class $y_q$ only considers the description generated specifically for it. This is **ensured by the summation condition** $a' \in \mathcal A (y_q)$ in Equation 5, which restricts the aggregation to the set of descriptions $\mathcal{A}(y_q)$ associated solely with class $y_q$.
> The computation thus aligns precisely with the intuition that class-specific descriptions should be used for each class.
>
> While Fig.3 visualizes the PRM matrix (which is sparse in fact, with most values being zero), the core class-specific computation is defined by Equation 5. A detailed explanation of this computation can be found in Appendix B.2. We hope this clarification resolves your concern.
>
> ---
>
> Thank you for your time! If you have any further questions, please feel free to ask—we’d be happy to provide more clarification. If our response has resolved your concerns, we hope you might reconsider your rating.

---

### Official Review · Reviewer_Gm1j · 2025-03-16

**Overall Recommendation:** 4

**Summary:**

This paper proposed the Decoupled Visual Prompts (DVP) to improve the limited learning capacity of the single VP. Explicit causes and unsupervised clusters are grouped with descriptions and decoupled-and-reweighted VPs are trained to improve the final performance. Experiments show better results, compared with the single VP.

**Claims And Evidence:**

Yes.

**Essential References Not Discussed:**

No.

**Experimental Designs Or Analyses:**

Yes.

Weaknesses or questions:
1. Please show the FPS and Number of Parameters of each method in Table 1.
2. Why the results of ATTRVR in this paper is different with that in the ATTRVR paper?

**Methods And Evaluation Criteria:**

Yes. The proposed methods are reasonable. Using the decoupled VP set should be better than using the single VP.

Weaknesses or questions:
1. Using the decoupled VP set would increase the runtime and computation resource. This should be analyzed through theory and experiments.
2. The probability reweighting matrix is computed by the bayes rule on the training set. But is it possible to train an MLP to replace the matrix?

**Other Comments Or Suggestions:**

None.

**Other Strengths And Weaknesses:**

None.

**Questions For Authors:**

None.

**Relation To Broader Scientific Literature:**

None.

**Theoretical Claims:**

No Theoretical Claims.

---

> ### Author Rebuttal · Authors · 2025-03-29
>
> M (Methods And Evaluation Criteria); E (Experimental Designs Or Analyses); R (Response)
>
> **M1/E1.** Increased computation resource caused by DVP
>
> **R-M1/E1:** Thank you for your suggestion! We have included the parameter numbers for various methods in the following table and will incorporate it into the main text in the next version.
>
> |                                |   VP  |   AR  | AttrVR | DVP-cse (ours) |   DVP-cls (ours)  | DVPlite (ours) |
> |:-:|:-:|:-:|:-:|:-:|:-:|:-:|
> |             params             | 0.07M | 0.04M |  0.04M |  0.12M  | 0.04-0.12M |  0.04M  |
> | Average accuracy over 11 tasks |  74.4 |  76.5 |  78.5  |   80.1  |    79.7    |   79.8  |
>
> Indeed, using DVP-cse or DVP-cls does increase the number of visual prompts, which will multiply the trainable parameter numbers.
>
> However, in response to your feedback, we proposed DVPlite--a decoupling method that keeps the parameter numbers the same as a single visual prompt--to test the effectiveness of our DVP-PRM framework. DVPlite decomposes the original padding-based visual reprogramming pattern into four separate components (top, bottom, left, and right), training them as independent visual prompts.
>
> |              | params   | Aircraft | Caltech | Cars | DTD  | ESAT | Flowers | Food | Pets | SUN  | UCF | Resisc | average |
> |-|-|-|-|-|-|-|-|-|-|-|-|-|-|
> | AttrVR-baseline |0.04M| 36.6     | 95.7    | 68.3 | 65.6 | 93.8 | 92.9    | 85.9 | 93.3 | 69.6 | 79  | 82.6   | 78.5    |
> | DVPlite  |0.04M       | 39.3     | 95.9    | 71.4 | 66.5 | 93.8 | 95.2    | 85.8 | 93.4 | 71.6 | 81  | 83.6   | 79.8    |
>
> As shown in the table above, DVPlite achieves performance comparable to DVP-cse or DVP-cls *without increasing the parameter numbers*, demonstrating its efficiency compared to the baseline method AttrVR.
>
> **M2.** The feasibility of using a fully connected layer
>
> **R-M2:** Thank you for your question! Replacing the Bayes rule computation of probabilistic reweighting matrix (PRM) with a single-layer neural network is theoretically feasible but presents two major issues:
>
> - *Significant increase in parameter count* – Assuming each class has $m$ descriptions and the total number of classes is $c$, the additional trainable parameters would be $mc^2$. For large-category tasks like SUN397, this results in two orders of magnitude more parameters than a visual prompt.
>
> - *Inability to Preserve PRM’s Sparsity* – As shown in Equation 5, PRM is a sparse matrix, meaning each class is only associated with its own descriptions, with zero weights for others. Using a fully connected layer to approximate this sparse matrix would likely yield suboptimal results.
>
> For these reasons, we compute PRM using the Bayes rule instead of a trainable linear layer.
>
> **E2.** Different results of the baseline method
>
> **R-E2:** Thank you for your question. For the baseline results, we directly used the results provided in the paper of AttrVR. The difference in the average value is because AttrVR uses 12 datasets, while our study focuses on 11 finer-grained downstream classification tasks to better figure out the functionality of each visual prompt, leading to a different average accuracy.
>
> ---
> Thank you again for your time!

---

> > ### Comment · Reviewer_Gm1j · 2025-04-02
> >
> > Thank you for your responses. They solved my questions and I am satisfied. Since I have already give the 4 score (accept), I will keep it. Hope your paper can be accepted.

---

> > > ### Author Response · Authors · 2025-04-02
> > >
> > > Dear Reviewer Gm1j,
> > >
> > > We are glad to hear that your concerns have been addressed! We very appreciate your valuable comments to us.
> > >
> > > Best regards,
> > >
> > > Authors of Submission 11510

---

### Official Review · Reviewer_MNe5 · 2025-03-17

**Overall Recommendation:** 2

**Summary:**

This paper proposes a visual reprogramming method for CLIP. The existing methods utilize a single visual prompt for all descriptions, but a single prompt may not enough to capture diverse aspects of class descriptions. The authors propose to have multiple visual prompt for each description (decouple), and integrate the outputs from each visual prompt with a probabilistic reweighting matrix. The proposed method was evaluated on various downstream classification tasks and obtain the improved results compared to other visual reprogramming methods.

**Claims And Evidence:**

Motivation: I agree that a single visual prompt is not appropriate to model all the descriptions for classification. However, Figure 2 may not appropriate to fully show the necessity of multi visual prompts because "Height" description score might be decreased due to the absence of visual cue for height in the example images. Even as human, we cannot say that the flower in the image are growing up to 4 or 5 feet tall. I recommend the authors to include the clearer sample that the description, which is clearly shown in the image, has decreased score after reprogramming.

Necessity of visual reprogramming: Visual programming is an interesting research topic, but I want to see the clear pros and cons compared to other PEFT methods (e.g., LoRA, visual prompt tuning, adapter, ...). I can easily come up with some cons of the visual reprogramming (e.g., fixed input resolution, requiring LLM heuristic description, ...) but not for pros of visual reprogramming. The clear comparison with other PEFT methods will be appreciated.

**Essential References Not Discussed:**

NA

**Experimental Designs Or Analyses:**

Baselines: I recommend the authors to include baselines from other family of PEFT methods, like visual prompt tuning, LoRA, adapters, etc, with presenting the number of learnable paramaters.

Fair comparison with baselines:
- To show the effectiveness of decoupling prompts, the reweighting technique should be applied on baselines in Table 1 (i.e., not just max/avg across all the descriptions, but learning the weight for each description).

- The authors should present the comparison of learnable params in Table 1. I guess the proposed method has more number of learnable parameters than other reprogramming methods because the proposed method has multiple visual prompts for each description.

**Methods And Evaluation Criteria:**

Yes, the proposed methods sound reasonable and the evaluation criteria is acceptable.

**Other Comments Or Suggestions:**

As done in visual prompt tuning paper [ref1], it would be great if the proposed method can also be applied on other visual tasks like segmentation, detection, etc.

[ref1] Visual Prompt Tuning

**Other Strengths And Weaknesses:**

Please see above

**Questions For Authors:**

It can be out of scope, but can this method be applied to multi-modal LLM?

**Relation To Broader Scientific Literature:**

The motivation is interesting, but as noted above, I don't see the advantages of visual reprogramming compared to other family of PEFT methods.

**Theoretical Claims:**

I checked the proofs for theoretical claims. I think the proof for Lemma 4.2 has some flaws. The main conclusion of this proof should show that "optimizing flexible models will result in lower empirical risks than identical models", but the authors use them as assumption (Lines 736-737). For more complete proofs, the authors should analytically proof the assumption as well.

---

> ### Author Rebuttal · Authors · 2025-03-29
>
> F (Feedback); R (Response)
>
> **F1.** Pros and cons of visual reprogramming (VR)
>
> **R1:** Key difference: VR applies transformations only to input and/or output, while other PEFTs adapt within internal architectures, leading to 3 pros of VR:
> - VR untouches any parameters and architecture, is applicable to *any* models, crucial for cases requiring entire model preservation to avoid catastrophic forgetting, copyright issues, or working with black-box APIs; in contrast, other PEFTs are architecture-dependent, e.g. VPT was sepcifically developed for ViT.
> - In VR, #trainable params depends only on *input data dims*, leading to lower training overhead compared with other PEFTs (e.g. LoRA, VPT) whose parameters scale with the pretrained model.
> - VR is orthogonal to other PEFTs and can be combined with them.
>
> So, VR has a unique position within PEFT the spectrum, offering complementary values to other methods.
>
> **F2.** Motivation experiment regarding the 'height' of flowers
>
> **R2:** We replaced the 'height' with 'color' descriptions (i.e., 'Artichoke petals exhibit a mix of soft purple, lavender, and green tones') and still observed similar results, confirming the reliability of our motivation. In the revision, we will refine results in Fig. 2 by replacing 'height's with 'color's.
>
> **F3.** Proof of Lemma 4.2
>
> **R3:** Thanks for pointing it out. We can analyze the hypothesis spaces induced by VR and DVP methods, which is the standard way to compare the potential performance of models with different capacities. Due to the character limit, we provide a proof sketch and will present the completed proof in the next version.
> 1. Define the hypothesis spaces representable by VR $\mathcal{H}\_{\rm VR}$ parameterized by $\delta$ and DVP $\mathcal{H}\_{\rm DVP}$ parameterized by $\Theta$, mapping inputs to outputs under fixed $\omega\_{\rm fix}$.
> 2. $\mathcal{H}\_{\rm VR} \subseteq \mathcal{H}\_{\rm DVP}$ as DVP has a larger parameter space and it specializes $\delta\_i$ for description partitions $\mathcal{A}\_i$, offering greater flexibility.
> 3. We conclude that optimization over the richer space allows DVP to find a function $f$ within $\mathcal{H}\_{\rm DVP}$ that fits training data $\mathcal{D}$ at least as well as the best function found within $\mathcal{H}\_{\rm VR}$.
>
> Thus, the minimum empirical risk $\hat{R}^{\rm dvp}\_{\mathcal{D}}$ searched over $\mathcal{H}\_{\rm DVP}$ must be less than or equal $\hat{R}^{\rm vr}\_{\mathcal{D}}$.
>
> **F4.** More experiments
>
> **R4:** We include more PEFTs (VPT, Adaptor) and mark the number of trainable parameters as reqeusted. Due to space limitations, we report the average accuracy across all 11 datasets. `DVPlite` is a new DVP variant with reduced parameters by decoupling the padding pattern of visual reprogramming (top, bottom, left, right) while maintaining the same number of parameters as the baseline.
> |      |   PEFT-VPT   | PEFT-Adaptor | VR-AttrVR | VR-DVP (ours) | VR-DVPlite (ours) |
> |-|-|-|-|-|-|
> |           params         |  0.04M  |  <0.40M  |  0.04M |    0.12M   |      0.04M      |
> |   RN50-CLIP (11 tasks)  | Invalid |   55.9  |  64.6  |    66.6    |        66.1       |
> | ViT16-CLIP (11   tasks) |   78.8  |   76.4  |  78.5  |    80.1    |       79.8      |
>
> Regarding the reweighting matrix (PRM): we did not apply it to baselines, as PRM is one of the contributions of *this study*. In the ablation study, we have evaluated the standalone effect of PRM to isolate its utility.
>
> We thus observe two key merits of our method:
> - Works with all architectures (RN, ViT-based), but VPT is incompatible with RN-based models.
> - Improve accuracy without necessarily increasing parameters.
>
> **F5.** Application in other tasks/multi-modal LLMs
>
> **R5:** While we follow previous VR studies on classification, the idea of DVP-PRM can extend to other tasks because CLIP learns general-purpose visual and text representations, suitable for tasks like segmentation and detection, and DVP *does not* fundamentally change the CLIP prediction pipeline.
> - In segmentation, DVP can motivate methods operating at the pixel/patch level with decoupled prompts specific to different semantic elements (boundaries, textures), while using PRM to generate dense prediction maps.
> - In detection, DVP can integrate into a two-stage detector, where decoupled prompts first generate region proposals focusing on objectness and scale variations, then region-specific PRM refines classification and localization.
>
> Besides, the workflow of DVP is similar on MLLM and CLIP, with the only difference being the use of instructions for downstream tasks.
>
> We appreciate these extensions are promising directions *beyond the scope* of this paper. In this paper, we focuse on extending single visual prompt capabilities and provide interpretability for reprogramming. Experiments on these applications can be explored in future work.
>
> ---
> Thank you for your time! If our response has resolved your concerns, we would appreciate a higher score.

---

> > ### Comment · Reviewer_MNe5 · 2025-04-07
> >
> > I appreciate the authors' response. However, I still have some concerns, which I believe this paper is not ready to be published. Therefore, I would keep my original rating for now.
> >
> > For R1,
> > - How VR is working with blackbox models? VR also needs gradient from model to optimize parameters
> > - LoRA is not dependent on architecture, and can work as plug-in module to avoid catastrophic forgetting
> > - VPT can also work as plug-in module
> > - To truly show the compatibility, the authors should show empirical evidence that "VR is orthogonal to other PEFTs and can be combined with them."
> >
> > For R3, "optimization over the richer space lead best function" is not 100% guaranteed in practice. I still the Lemma 4.2 has a flaw by using the conclusion as assumption
> >
> > For R4, for more complete comparison, the authors can include LoRA with comparable # of parameters by adjusting rank values.

---

> > > ### Author Response · Authors · 2025-04-08
> > >
> > > R1: We'd like to clarify a fundamental point about our paper's positioning and then address the specific concerns regarding evidence.
> > >
> > > - **Research direction and contribution**. Our paper **has never claimed** that VR is superior to other PEFT methods. Rather, VR **is the problem setting** within which our work operates. The contribution of this paper is in understanding and improving current VR methods for CLIP through our decoupled visual prompts strategy, **not** in establishing VR itself as preferable to alternatives like LoRA or VPT.
> > > - **Value of VR**. As we previously replied, VR has a unique position because it only relies on input and output space modifications. While VR has some limitations, exploring and improving VR methods contributes valuable knowledge to the broader field of PEFT (this is also why this paper comes out). We **respectfully disagree** with the reviewer's opinion that VR has no pros; and importantly we'd like to highlight the importance of understanding the full spectrum of PEFT approaches - including VR - will enrich the field's capabilities, because no single method is universally superior. It is thus **unfair** to arbitrarily dismiss the value of an active research area like VR.
> > >
> > > Response to evidence concerns and clarification of misunderstanding:
> > > - **black-box optimization**. The black-box VR **has been successfully explored** in existing studies, such as BlackVIP [1] and BAR [2]. These methods demonstrate how VR can be applied **without direct gradient access**.
> > > - **Empirical evidence of orthogonality**. There is also existing evidence confirming that VR can indeed be combined with LoRA [3] for complementary improvements.
> > > - **Architecture independence**. VR is inherently and universally applicable across **any architecture** by design; in contrast, while we understand that LoRA and VPT could work as plug-in methods, they **were initially proposed for transformer-based models**.
> > > - Importantly, we **didn't claim** that VR is the only method that can prevent catastrophic forgetting.
> > >
> > >
> > > [1]. Black-box visual prompting for robust transfer learning. CVPR 2023.
> > >
> > > [2]. Transfer learning without knowing: Reprogramming black-box machine learning models with scarce data and limited resources. ICML 2020
> > >
> > > [3]. Sample-specific masks for visual reprogramming-based prompting. ICML 2024.
> > >
> > >
> > > R3: We'd like to clarify a potential misunderstanding: we **didn't state** that "optimization over the richer space lead best function", but we were trying to compare the **best achievable function within** $\mathcal{H}\_{\rm VR}$ against **best achievable function within** $\mathcal{H}\_{\rm DVP}$. Our conclusion is based on **comparison between hypothesis spaces**, following directly from fundamental principles in statistical learning theory:
> > > - **Hypothesis Space Containment**: We can formally derive that $\mathcal{H}\_{\rm VR}$ is **strictly contained** within $\mathcal{H}\_{\rm DVP}$.
> > > - **Approximation Error**: It quantifies the best possible performance gap between the true underlying function and the best predictor within a given hypothesis class.
> > > - **Approximation Error Guarantee**: Given this containment relationship, DVP is **formally guaranteed** to have a lower or equal approximation error than VR.
> > >
> > > Thus, this **is not an assumption** but a mathematical property that follows from set inclusion and the definition of approximation error. The inequality holds **regardless of optimization challenges** in practice - it refers to the **theoretical minimum achievable** within each hypothesis space.
> > >
> > > R4: As with our reply to R1, we respectfully clarify that the core contribution of our paper is to understand and improve VR, **rather than** comparing different PEFT approaches across **different problem settings**.
> > >
> > > The table we provided in the rebuttal serves two specific purposes:
> > > 1. To demonstrate that existing VR methods (including our DVP) work consistently across different architectures (both RN and ViT-based), unlike some PEFT methods (e.g., VPT) which might have **architectural constraints by default**.
> > > 2. To show that DVP's improvements to VR **don't necessarily require parameter increases**, as evidenced by `DVP-lite` achieving gains while maintaining parameter parity with baseline VR methods.
> > >
> > > Adding LoRA into comparison against VR would be **tangential to our research focus and problem settings**
> > > - LoRA and VR are **not competitive** methods, but offer complementary benefits (per our reply to R1)
> > > - Our analysis, the DVP-PRM framework is specifically designed for VR methods. These contributions are an integral component of our approach, not an independent technique that should be applied to baselines.
> > >
> > > We believe the current experimental setup directly supports our central claims about improving VR through decoupling and reweighting, while the requested cross-paradigm comparisons would shift focus away from the paper's core contributions and potentially confuse readers about its intended scope.

---

### Decision · Program_Chairs · 2025-05-01

**Decision:**

Accept (poster)

**Comment:**

The authors propose a Decoupled Visual Prompts algorithm with reweighting strategy, to improve the VR method, as an alternative of existing PET methods.   The authors also find the paper well-structured and the proposed algorithm is supported by the experiments.  There are some discrepancy during the rebuttal, and the authors address them well. Please include all of the additional results and clarification for the final version.